# Bird population declines and species turnover are changing the acoustic properties of spring soundscapes

C. A. Morrison[1], A. Auniņš [2,3], Z. Benkő[4,5], L. Brotons[6,7,8], T. Chodkiewicz [9,10], P. Chylarecki [9], V. Escandell[11], D. P. Eskildsen[12], A. Gamero[13], S. Herrando [7,14], F. Jiguet [15], J. A. Kålås [16], J. Kamp[17,18], A. Klvaňová [13], P. Kmecl [19], A. Lehikoinen [20], Å. Lindström [21], C. Moshøj [12], D. G. Noble[22], I. J. Øien [23], J-Y. Paquet [24], J. Reif[25,26], T. Sattler [27], B. S. Seaman [28], N. Teufelbauer [28], S. Trautmann [18], C. A. M. van Turnhout[29,30], P. Voříšek [13,26] & S. J. Butler [1✉]

Natural sounds, and bird song in particular, play a key role in building and maintaining our connection with nature, but widespread declines in bird populations mean that the acoustic properties of natural soundscapes may be changing. Using data-driven reconstructions of soundscapes in lieu of historical recordings, here we quantify changes in soundscape characteristics at more than 200,000 sites across North America and Europe. We integrate citizen science bird monitoring data with recordings of individual species to reveal a pervasive loss of acoustic diversity and intensity of soundscapes across both continents over the past 25 years, driven by changes in species richness and abundance. These results suggest that one of the fundamental pathways through which humans engage with nature is in chronic decline, with potentially widespread implications for human health and well-being.

[1] School of Biological Sciences, University of East Anglia, Norwich, UK. [2] Faculty of Biology, University of Latvia, Jelgavas iela 1, Riga LV-1004, Latvia. [3] Latvian Ornithological Society, Skolas iela 3, Riga LV-1010, Latvia. [4] Romanian Ornithological Society/BirdLife Romania, Cluj-Napoca, Romania. [5] Evolutionary Ecology Group, Hungarian Department of Biology and Ecology, Babeș-Bolyai University, Cluj-Napoca, Romania. [6] InForest JRU (CTFC-CREAF), Solsona 25280, Spain. [7] CREAF, Cerdanyola del Vallès, 08193 Barcelona, Spain. [8] CSIC, Cerdanyola del Vallès, 08193 Barcelona, Spain. [9] Museum and Institute of Zoology, Polish Academy of Sciences, Wilcza 64, 00-679 Warszawa, Poland. [10] Polish Society for the Protection of Birds (OTOP), ul. Odrowaza 24, 05-270 Marki, Poland. [11] Sociedad Española de Ornitología (SEO/BirdLife), Madrid, Spain. [12] Dansk Ornitologisk Forening, BirdLife Denmark, Vesterbrogade 138-140, DK-1620 København V, Denmark. [13] European Bird Census Council-Czech Society for Ornithology, Na Bělidle 34, 15000 Prague 5, Czechia. [14] European Bird Census Council–Catalan Ornithological Institute, Natural History Museum of Barcelona, Plaça Leonardo da Vinci 4-5, 08019 Barcelona, Catalonia, Spain. [15] Centre d'Ecologie et des Sciences de la Conservation, UMR7204 MNHN-CNRS-SU, Paris, France. [16] Norwegian Institute for Nature Research, P.O. Box 5685Torgarden, NO-7485 Trondheim, Norway. [17] University of Göttingen, Department of Conservation Science, Bürgerstr. 50, 37073 Göttingen, Germany. [18] Dachverband Deutscher Avifaunisten (DDA), An den Speichern 2, 48157 Münster, Germany. [19] DOPPS - BirdLife Slovenia, Tržaška cesta 2, SI-1000 Ljubljana, Slovenia. [20] Finnish Museum of Natural History, FI-00014 University of Helsinki, P.O. Box 17 Helsinki, Finland. [21] Biodiversity Unit, Department of Biology, Lund University, Ecology Building, S-223 62 Lund, Sweden. [22] British Trust for Ornithology, The Nunnery, Thetford, Norfolk IP24 2PU, UK. [23] NOF-BirdLife Norway, Sandgata 30 B, NO-7012 Trondheim, Norway. [24] Natagora, Département Études, Traverse des Muses 1, B-5000 Namur, Belgium. [25] Institute for Environmental Studies, Faculty of Science, Charles University in Prague, Prague, Czechia. [26] Department of Zoology and Laboratory of Ornithology, Faculty of Science, Palacký University Olomouc, 17 Listopadu 50, 771 43 Olomouc, Czechia. [27] Swiss Ornithological Institute, Seerose 1, 6204 Sempach, Switzerland. [28] BirdLife Österreich, Museumsplatz 1/10/8, A-1070 Wien, Austria. [29] Sovon Dutch Centre for Field Ornithology, P.O. Box 6521, 6503 GA Nijmegen, Netherlands. [30] Department of Animal Ecology and Ecophysiology, Institute for Water and Wetland Research, Radboud University, P.O. Box 9010, 6500 GL Nijmegen, Netherlands. ✉email: simon.j.butler@uea.ac.uk

O ver half the world's population now live in cities[1]. Rapid urbanisation, along with increasingly sedentary lifestyles associated with a rise in electronic media, changing social norms, and shifting perceptions around outside play[2–4], are reducing people's opportunities for direct contact with the natural environment. This so-called extinction of experience[5] is driving a growing human-nature disconnect, with negative impacts on physical health, cognitive ability and psychological well-being[6–10]. The COVID-19 pandemic has highlighted this issue, both in terms of the detrimental impacts on mental health due to local and national lockdowns imposed by governments and the wide-spread recognition of the benefits of engaging with nature during this period[11,12]. Global biodiversity loss[13] is also likely to be driving a dilution of experience, whereby the quality of those interactions with nature which do still occur is also being reduced[14] but we do not yet know the extent of such changes.

Sound confers a sense of place and is a key pathway for engaging with, and benefitting from, nature[15]. Indeed, since Rachel Carson's (1962) classic book "Silent Spring", nature's sounds have been inextricably linked to perceptions of environ-mental quality[16], and the maintenance of natural soundscape integrity is increasingly being incorporated into conservation policy and action[17]. Birds are a major contributor to natural soundscapes[18] and bird song, and song diversity in particular, plays an important role in defining the quality of nature experiences[15,19–21]. Widespread reductions in both avian abundance[22] and species richness[23], alongside increased biotic homogenisation[24], are therefore likely to be impacting the acoustic properties of natural soundscapes and potentially redu-cing the quality of nature contact experiences[25]. Indeed, given that people predominantly hear, rather than see, birds[26,27], reductions in the quality of natural soundscapes are likely to be the mechanism through which the impact of ongoing population declines is most keenly felt by the general public. However, the relationship between changes in avian community structure and the acoustic properties of natural soundscapes is nuanced and non-linear[28]—the loss of a warbler species with a rich, complex song is likely to have a greater impact on soundscape char-acteristics than the loss of a raucous corvid or gull species, but this will depend on how many, and which, other species are present. The implications of biodiversity loss for local soundscape characteristics therefore cannot be directly predicted from count data alone.

Here we combine annual systematic bird count data from North American Breeding Bird Survey (NA-BBS) and Pan-European Common Bird Monitoring Scheme (PECBMS) sites with recordings of individual bird species, downloaded from an online database (www.xeno-canto.org), to reconstruct historical soundscapes at over 200,000 locations across the two continents over the past 25 years. Taking the first species listed in a site-year count data file, a 25 s sound file for that species was inserted at a random time point in an initially empty 5 min sound file. Play-back volume was randomly sampled from a uniform distribution to represent varying proximity of individual birds to the surveyor. This process was repeated as many times as there were indivi-duals of the first species counted, and then for all individuals of all other species in that site-year count data file, to build a single, composite representation of the local soundscape for the year when those count data were collected. This process was repeated for all site-year count data files, so that separate soundscapes were constructed for every site in every year it was surveyed. We employed a systematic protocol for soundscape construction, applying the same rules for translating survey data into sounds-cape contribution across all species, because data on vocalisation frequency (how often an individual vocalises) and duration (how long each vocalisation event lasts) are not available for most

species included in our analyses. However, while standardised in length, the 25 s sound files used to represent an individual of a given species did comprise interspersed periods of vocalisation and silence, and therefore captured the inherent variation in song or call structure and pattern of delivery between species to some extent.

The acoustic characteristics of these reconstructed soundscapes were then quantified using four indices designed to capture the distribution of acoustic energy across frequencies and time[29] and to reflect the richness (Acoustic Diversity Index: ADI[30]), evenness (Acoustic Evenness Index: AEI[30]), amplitude (Bioacoustic Index: BI[31]) and heterogeneity (Acoustic Entropy: H[32]) of each soundscape. These acoustic indices are broadly correlated with avian species richness and abundance[30–33] but are fundamentally driven by song complexity and diversity across contributing species. They therefore describe the key factors predicted to underpin public perceptions of the quality of their nature experiences[15,19–21], with lower values of ADI, BI and H, and higher values of AEI, reflecting reduced acoustic diversity and intensity. These indices respond in a similar way when applied to constructed soundscapes generated from simulated communities varying in species richness and abundance, with both increasing abundance and species richness leading to increases in ADI, BI and H and a decrease in AEI (Figs. 1, 2; Tables 1, 2). These relationships are not linear, with the rate of increase in BI and H with increasing abundance lower at higher species richness (Fig. 2; Table 2) and each index becoming less sensitive to changes in community structure as soundscapes become more saturated.

Acoustic indices have been used to explore diel and annual patterns in soundscape structure[34,35] and to characterise differ-ences in soundscapes across habitats and landscapes[33,36,37]. However, evidence of changes in soundscape characteristics over longer time periods is currently lacking because of a scarcity in historical soundscape recordings. By reconstructing soundscapes from large-scale bird monitoring datasets and archived recordings of individual species, both predominantly generated by citizen scientists, we are able to explore changes in soundscape quality at sites across North America and Europe over recent decades. We reveal a chronic deterioration in soundscape quality, defined as a reduction in acoustic diversity and/or intensity, across both continents. Our analyses suggest that changes in the com-position, diversity and abundance of bird communities are all likely to have contributed to this. Ongoing declines in bird populations[13,22] are expected to cause further reductions in soundscape quality and, by extension, a continued dilution of the nature contact experience.

## Results and discussion

We identify patterns of significant and broadly parallel declines in ADI, BI and H across both continents since the late 1990s, and a significant increase in AEI in North America over the same period (Fig. 3; Table 3). These changes suggest that natural soundscapes have, overall, become both more homogeneous and quieter. Within these general patterns of reduced soundscape quality, there was substantial site-level variation, with local declines and increases in all four indices occurring across each continent (Fig. 4, Supplementary Fig. 1), while larger-scale geo-graphical patterns in the rates of change in each index are also evident (Supplementary Tables 1, 2). For example, reductions in acoustic diversity (signalled by decreases in ADI and increases in AEI) have been greatest in the North and West of both continents (Supplementary Figs. 2a–d, 3a–d), while soundscape intensity, as measured by BI, has declined most in more northern and eastern areas of North America but shows no spatial pattern across

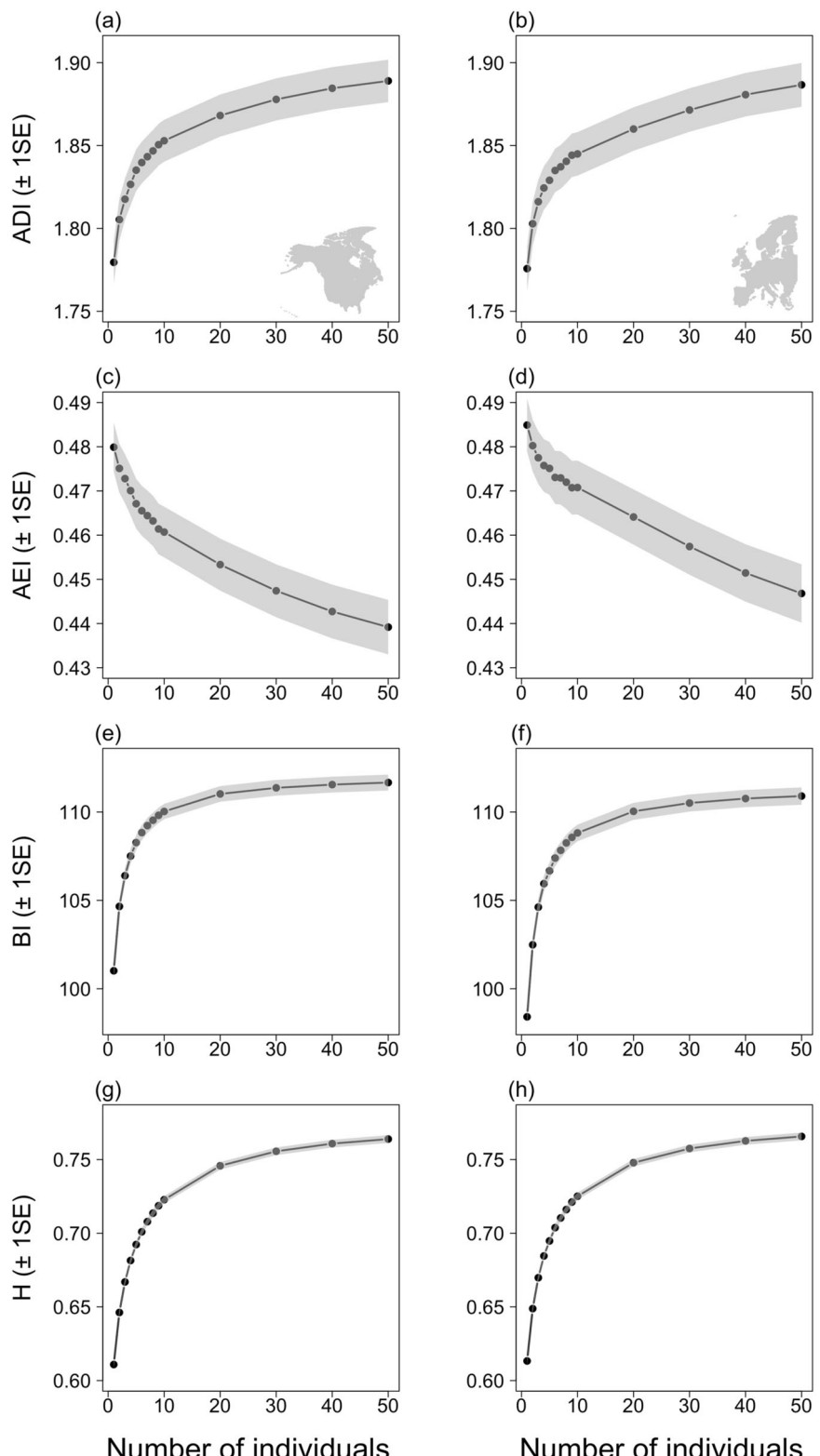

**Fig. 1 Responses of acoustic indices to changes in abundance in simulated communities.** The association between the Acoustic Diversity Index, ADI (**a**, **b**), Acoustic Evenness Index, AEI (**c**, **d**), Bioacoustic Index, BI (**e**, **f**) and Acoustic Entropy, H (**g**, **h**) of constructed soundscapes and the number of individuals of a single species contributing to that soundscape for North American (left column) and European (right column) species. Shaded areas indicate ±1 standard error.

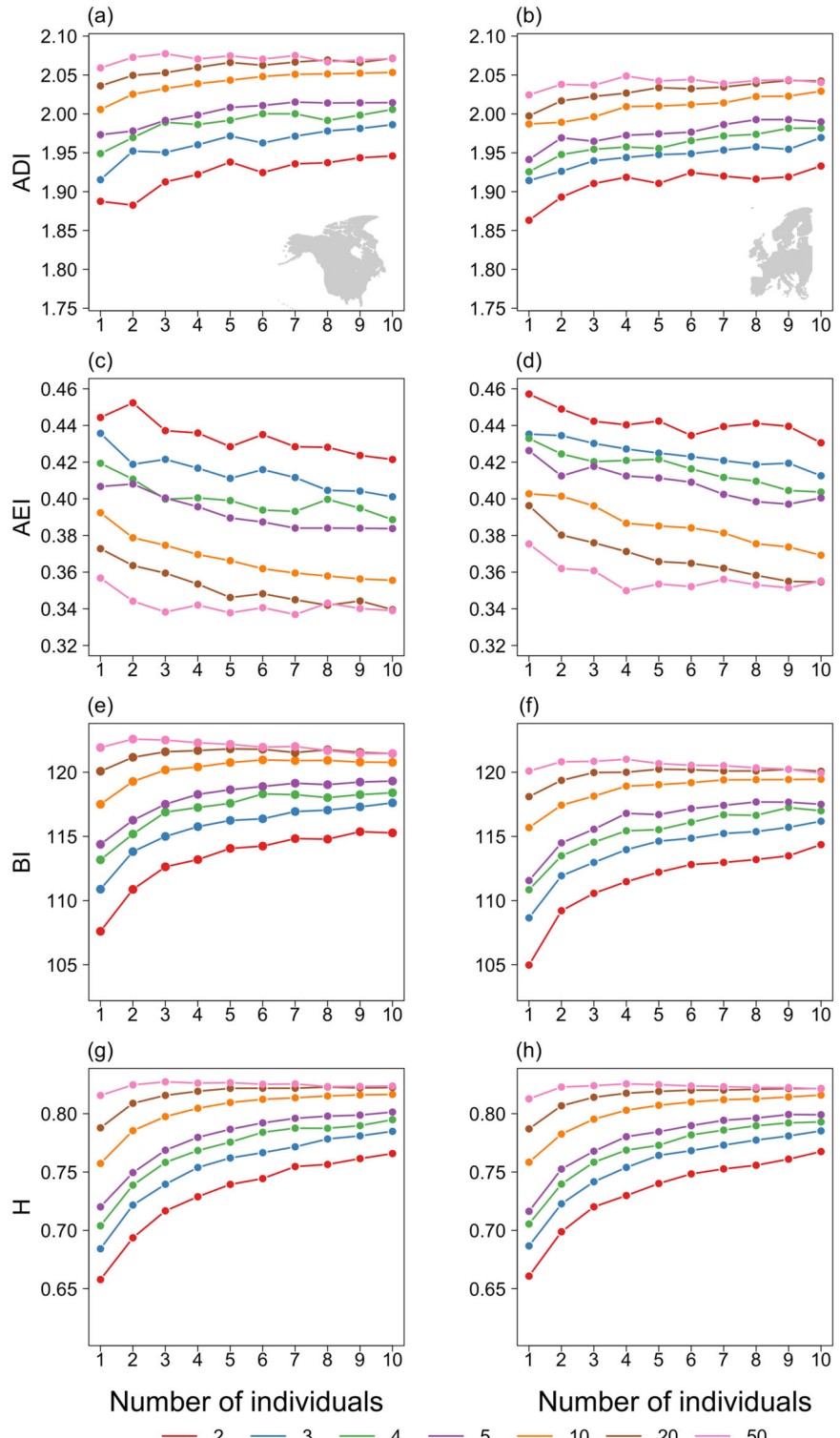

**Fig. 2 Responses of acoustic indices to changes in simulated community structure.** The association between the Acoustic Diversity Index, ADI (**a**, **b**), Acoustic Evenness Index, AEI (**c**, **d**), Bioacoustic Index, BI (**e**, **f**) and Acoustic Entropy, H (**g**, **h**) of constructed soundscapes and the number of individuals and species contributing to that soundscape for North American (left column) and European (right column) species. Colours indicate the number of species in the community.

Europe (Supplementary Figs. 2e,f, 3e,f). In contrast, while H has also decreased more in eastern North America, it has also decreased slightly more in the south than in the north (Supplementary Fig. 2g,h). In Europe, H has decreased in northern and western areas but increased slightly towards the south and east (Supplementary Fig. 3g,h).

Local soundscape dynamics are likely to be underpinned by multiple and interacting processes, operating at regional, biome and local levels, which influence species richness and abundance[38,39], taxonomic, functional and phylogenetic diversity[40,41], and the rate and direction of change in community composition[22,42–45]. Overall, there has been a significant decline

**Table 1 Results of GLMs of the association between the acoustic properties of reconstructed soundscapes and the number of individuals (1–10) of a single species present in a simulated community and contributing to that soundscape.**

| | Fixed effects | North America | | | Europe | | |
|---|---|---|---|---|---|---|---|
| | | Estimate (SE) | t | p | Estimate (SE) | t | p |
| (a) ADI | (Intercept) | 1.7880 (0.0021) | 832.12 | <0.001 | 1.7880 (0.0073) | 246.01 | <0.001 |
| | Log(Individuals) | 0.0270 (0.0009) | 29.49 | <0.001 | 0.0269 (0.0031) | 8.72 | <0.001 |
| (b) AEI | (Intercept) | 0.4836 (0.0011) | 456.74 | <0.001 | 0.4836 (0.0033) | 146.15 | <0.001 |
| | Log(Individuals) | −0.0106 (0.0005) | −23.45 | <0.001 | −0.0106 (0.0014) | −7.50 | <0.001 |
| (c) BI | (Intercept) | 103.52 (0.65) | 160.19 | <0.001 | 103.5215 (0.2495) | 414.89 | <0.001 |
| | Log(Individuals) | 2.46 (0.28) | 8.93 | <0.001 | 2.4600 (0.1063) | 23.14 | <0.001 |
| (d) H | (Intercept) | 0.6255 (0.0047) | 133.1 | <0.001 | 0.6255 (0.0014) | 445.80 | <0.001 |
| | Log(Individuals) | 0.0389 (0.0020) | 19.4 | <0.001 | 0.03891 (0.0006) | 64.98 | <0.001 |

(a) Acoustic Diversity Index (ADI), (b) Acoustic Evenness Index (AEI), (c) Bioacoustic Index (BI) and (d) Acoustic Entropy (H). Separate models for North American (left column) and European (right column) species presented.

**Table 2 Results of GLMs of the association between the acoustic properties of reconstructed soundscapes and the number of individuals (1–10) and species (2,3,4,5,10,20,50) present in a simulated community and contributing to that soundscape.**

| | Fixed effects | North America | | | Europe | | |
|---|---|---|---|---|---|---|---|
| | | Estimate (SE) | t | p | Estimate (SE) | t | p |
| (a) ADI | (Intercept) | 1.866 (0.011) | 175.12 | <0.001 | 1.856 (0.008) | 222.38 | <0.001 |
| | Log(Individuals) | 0.034 (0.006) | 5.35 | <0.001 | 0.028 (0.005) | 5.66 | <0.001 |
| | Log(Species) | 0.056 (0.005) | 11.85 | <0.001 | 0.047 (0.004) | 12.57 | <0.001 |
| | Log(Individuals) *Log(species) | −0.007 (0.003) | −2.61 | 0.011 | −0.005 (0.002) | −2.13 | 0.037 |
| (b) AEI | (Intercept) | 0.461 (0.003) | 148.60 | <0.001 | 0.472 (0.002) | 203.68 | <0.001 |
| | Log(Individuals) | −0.012 (0.001) | −8.30 | <0.001 | −0.012 (0.001) | −10.88 | <0.001 |
| | Log(Species) | −0.029 (0.001) | −29.67 | <0.001 | −0.027 (0.001) | −37.18 | <0.001 |
| (c) BI | (Intercept) | 107.09 (0.662) | 161.84 | <0.001 | 104.53 (0.651) | 160.46 | <0.001 |
| | Log(Individuals) | 3.913 (0.398) | 9.83 | <0.001 | 4.367 (0.392) | 11.15 | <0.001 |
| | Log(Species) | 4.299 (0.293) | 14.69 | <0.001 | 4.49 (0.288) | 15.59 | <0.001 |
| | Log(Individuals) *Log(species) | −1.103 (0.176) | −6.27 | <0.001 | −1.170 (0.173) | −6.75 | <0.001 |
| (d) H | (Intercept) | 0.639 (0.006) | 102.04 | <0.001 | 0.643 (0.006) | 108.93 | <0.001 |
| | Log(Individuals) | 0.058 (0.004) | 15.33 | <0.001 | 0.056 (0.004) | 15.77 | <0.001 |
| | Log(Species) | 0.049 (0.003) | 18.05 | <0.001 | 0.050 (0.003) | 18.39 | <0.001 |
| | Log(Individuals) *Log(species) | −0.014 (0.002) | −8.59 | <0.001 | −0.014 (0.002) | −8.73 | <0.001 |

(a) Acoustic Diversity Index (ADI), (b) Acoustic Evenness Index (AEI), (c) Bioacoustic Index (BI) and (d) Acoustic Entropy (H). Separate models for North American (left column) and European (right column) species presented. Only significant interactions are retained.

in both the total number of species and individuals counted during NA-BBS surveys and in the total number of individuals counted during PECBMS surveys over the past 25 years (Supplementary Fig. 4; Supplementary Table 3). Importantly, there were strong positive relationships between site-level trends in ADI, BI and H and site-level trends in both species richness and the total number of individuals counted, with equivalent negative relationships for site-level trends in AEI (Table 4); sites that have experienced greater declines in either total abundance and/or species richness also show greater declines in acoustic diversity and intensity while sites, where total abundance and/or species richness has increased, tend to show increases in these characteristics (Fig. 5).

There were generally strong correlations in the trends of the four acoustic indices at each site, positive between ADI, H and BI, and negative between AEI and the other three (Supplementary Table 4). However, these patterns were not universal, with all potential combinations of increases or decreases in each index observed (Supplementary Fig. 5). Furthermore, there was substantial variation in the scale of change in a given acoustic index

for any given change in species richness or abundance (Fig. 5). Thus, while soundscape dynamics are fundamentally driven by changes in community structure, shifts in soundscape characteristics arising from changes in species composition and/or abundance over time are both multi-dimensional and context-dependent; measures of acoustic richness, evenness, amplitude and heterogeneity respond independently according to both initial community structure and how the call and song characteristics of constituent species compare[28]. Additional analyses are needed to understand the drivers of both this local site-level variation and the broader geographic patterns in soundscape dynamics, as well as the specific influence of changes in the abundance or occurrence of individual species.

While predominantly driven by community composition, it is important to recognise that the acoustic properties of reconstructed soundscapes could be influenced by methodological decisions applied during the construction process. For example, the ratio of individual sound file duration to total soundscape length will influence the degree of overlap between the calls and songs of individuals, while the probability distribution from

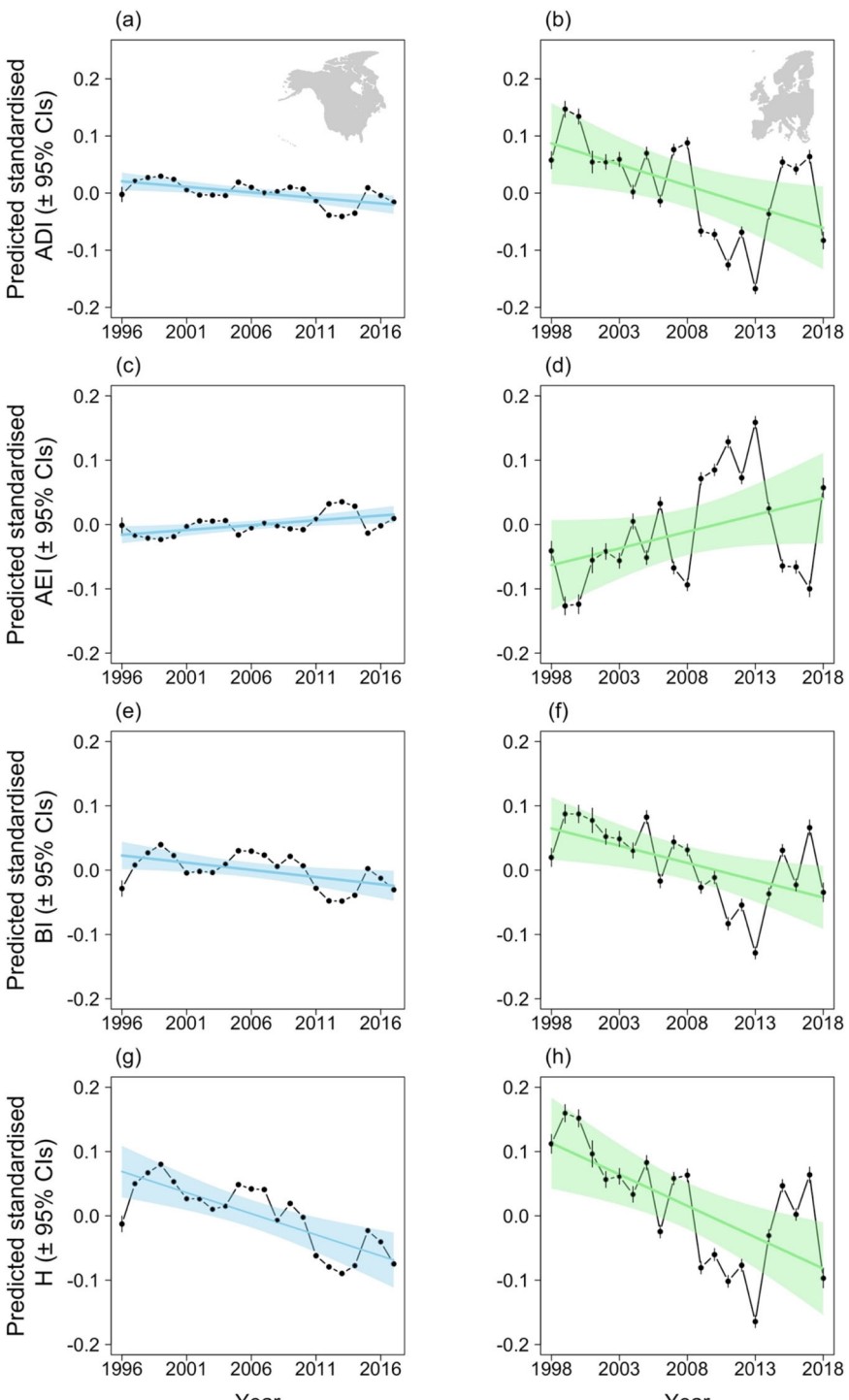

**Fig. 3 Temporal trends in acoustic indices.** Predicted annual variation in Acoustic Diversity Index, ADI (**a**, **b**), Acoustic Evenness Index, AEI (**c**, **d**), Bioacoustic Index, BI (**e**, **f**) and Acoustic Entropy, H (**g**, **h**) in North America (left column) between 1996 and 2017 and in Europe (right column) between 1998 and 2018. Blue (North America) and green (Europe) lines show the predicted trends from GLMMs (Table 3); shaded areas indicate 95% confidence intervals. Points show predicted annual values from GLMMs with the identical structure as those in Table 3 but with year fitted as a categorical rather than a continuous variable, vertical lines indicate the 95% confidence intervals. Annual values of each acoustic index were standardised at the site-level prior to analyses.

which playback volume is sampled will determine the relative proportion of near and far individuals in the soundscape. As a consequence, these methodological decisions influence the distribution of acoustic energy within each reconstructed soundscape and thus the absolute values of each acoustic metric[29]. To explore the implications of these decisions for detecting changes

in soundscape characteristics over time, we constructed soundscapes for 1000 simulated communities containing ten randomly selected species that each declined from 10 to 5 individuals over a 6-year period. For each community, we constructed soundscapes using four alternative approaches that altered the ratio of individual sound file duration to total soundscape length or the

**Table 3 Results of GLMMs of the variation in the acoustic properties of soundscapes.**

| | Fixed effects | North America | | | | Europe | | | |
|---|---|---|---|---|---|---|---|---|---|
| | | Estimate (SE) | $\chi^2$ | DF | p | Estimate (SE) | $\chi^2$ | DF | p |
| (a) ADI | Latitude | 0.00002 (0.00088) | 0.03 | 1 | 0.856 | −0.0004 (0.0005) | 0.78 | 1 | 0.376 |
| | Longitude | 0.00002 (0.00035) | 0.19 | 1 | 0.659 | 0.0005 (0.0003) | 3.57 | 1 | 0.059 |
| | Year | −0.00198 (0.00068) | 8.46 | 1 | 0.004 | −0.0073 (0.0030) | 5.98 | 1 | 0.014 |
| (b) AEI | Latitude | −0.00001 (0.00009) | 0.01 | 1 | 0.949 | −0.0005 (0.0005) | 1.25 | 1 | 0.263 |
| | Longitude | −0.00009 (0.00003) | 0.08 | 1 | 0.782 | −0.0005 (0.0003) | 2.84 | 1 | 0.092 |
| | Year | 0.00144 (0.00057) | 6.47 | 1 | 0.011 | 0.0053 (0.0030) | 3.08 | 1 | 0.079 |
| (c) BI | Latitude | 0.00004 (0.00009) | 0.16 | 1 | 0.691 | −0.0002 (0.0005) | 0.25 | 1 | 0.621 |
| | Longitude | 0.00002 (0.00003) | 0.44 | 1 | 0.508 | 0.0004 (0.0003) | 2.26 | 1 | 0.132 |
| | Year | −0.00217 (0.00088) | 6.08 | 1 | 0.014 | −0.0056 (0.0021) | 6.91 | 1 | 0.009 |
| (d) H | Latitude | 0.00010 (0.00009) | 1.38 | 1 | 0.239 | −0.0004 (0.0005) | 0.86 | 1 | 0.353 |
| | Longitude | 0.00006 (0.00004) | 3.19 | 1 | 0.074 | 0.0006 (0.0003) | 4.65 | 1 | 0.031 |
| | Year | −0.00625 (0.00175) | 12.79 | 1 | <0.001 | −0.0095 (0.0030) | 10.01 | 1 | 0.002 |

Annual values for each acoustic index were standardised at the site-level prior to analyses.
(a) Acoustic Diversity Index (ADI), (b) Acoustic Evenness Index (AEI), (c) Bioacoustic Index (BI) and (d) Acoustic Entropy (H) in 202737 NA-BBS sites across North America between 1996 and 2017 and in 16524 PECBMS sites across Europe between 1998 and 2018.

proportion of near to far individuals. While the methodological decisions applied during soundscape construction influenced the absolute values of the four acoustic indices for a given community, it did not influence the relative impact of changes in community composition on the acoustic indices (Supplementary Fig. 6). Given our focus here is on temporal trends in soundscape characteristics, rather than absolute values of each acoustic index, and that our analyses are based on changes in standardised site-level measures, we believe the temporal and spatial patterns in soundscape characteristics reported here are robust to the soundscape construction rules applied.

Natural soundscapes are under ever-increasing pressure from global biodiversity loss and our results reveal a chronic deterioration in soundscape quality across North America and Europe over recent decades. Although we focus here on birds as the main contributors to natural soundscapes, it is likely that the reduction in quality has been even greater, given parallel declines in many other taxonomic groups that contribute to soundscapes[46,47]. Furthermore, pervasive increases in anthropogenic noise[48] and other sensory pollutants[49] are also diluting the nature contact experience. For example, as well as directly impacting human behaviour and well-being[50], noise pollution impairs our capacity to perceive natural sounds[51] and can limit the acoustic diversity of soundscapes by constraining the bandwidth within which birds sing[52,53].

A scarcity of historical recordings means any assessment of changes in natural soundscape characteristics over longer time periods is vulnerable to the impacts of shifting baseline syndrome[54], as future soundscapes can only be compared to the potentially already degraded soundscapes of today. Reconstructing soundscapes from species' records and count data avoids this problem and allows changes in local soundscape characteristics to be explored at spatial scales not possible using field recordings. This approach could also be used to forecast future soundscapes based on projected species' range shifts under environmental change scenarios. However, we strongly advocate for the increased collection and systematic curation of soundscape field recordings from across habitats and environmental gradients to capture all facets of soundscape dynamics, such as changes in anthropogenic noise and vocalisation behaviour across taxonomic groups, not currently integrated into our reconstructions. The rapid increase in autonomous sound recording tools and their widespread use could be harnessed both to launch standardized soundscape monitoring schemes, and to collect soundscape recordings in less structured citizen science databases[55]. Such recordings could also be used to derive the vocalisation frequency and duration data needed to further enhance soundscape reconstructions by encoding species-specific insertion criteria in place of the systematic protocol (one individual equals one 25 s sound file) currently applied across all species.

Although visual, auditory, and olfactory senses are all important modalities characterising the nature contact experience[19,20], sound is a defining feature[15]. Our analyses of reconstructed soundscapes reveal previously undocumented changes in the acoustic properties of soundscapes across North America and Europe over the past few decades that signal a reduction in soundscape quality and imply an ongoing dilution of experience associated with nature interactions. While we expect these changes to be evident throughout the year, they are likely to be most pronounced during spring, when birds are most vocally active. Better understanding of exposure to changes in soundscape quality, by mapping them onto spatial patterns of human population density and locations at which nature is accessed, and of the specific soundscape characteristics that support and enhance the nature contact experience[15], is now needed to fully appreciate the implications for health and well-being[56]. Reduced nature connectedness may also be contributing to the global environmental crisis, as there is evidence it can lead to reductions in pro-environmental behaviour[5,57,58]. The potential for declining soundscape quality to contribute to a negative feedback loop, whereby a decline in the quality of nature contact experiences leads to reduced advocacy and financial support for conservation actions, and thus to further environmental degradation[7], must also be recognised and addressed. Conservation policy and action need to ensure the protection and recovery of high-quality natural soundscapes to prevent chronic, pervasive deterioration and associated impacts on nature connectedness and health and well-being.

## Methods

**Bird data**. *North America*: we used annual bird count data collated under the North American Breeding Bird Survey (NA-BBS): https://www.pwrc.usgs.gov/bbs/) from 1996 to 2017. NA-BBS survey routes, consisting of 50 survey points (hereafter sites) evenly distributed over ~24.5 miles, are distributed across the United States and Canada and are usually surveyed in June. At each site, skilled volunteers conduct a three-minute point count, recording all birds seen or heard within a 400-m radius[59].

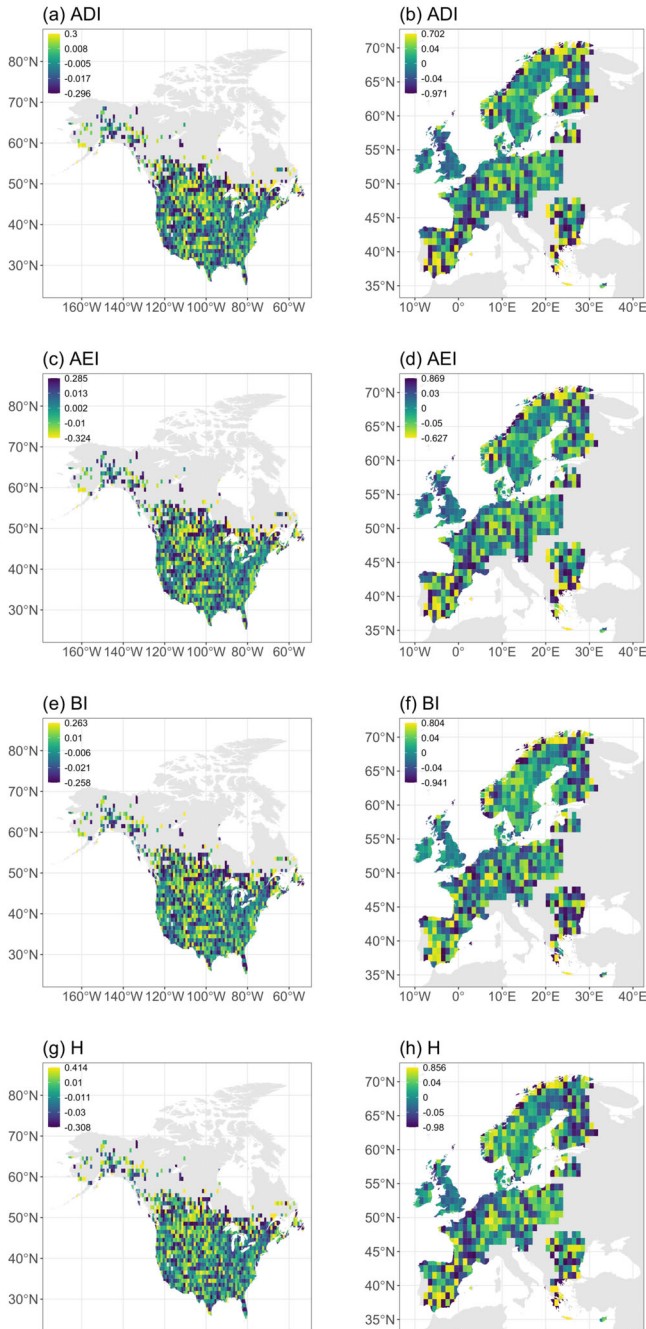

**Fig. 4 Spatial variation in acoustic index trends.** Mean site-level trend in Acoustic Diversity Index, ADI (**a**, **b**), Acoustic Evenness Index, AEI (**c**, **d**), Bioacoustic Index, BI (**e**, **f**) and Acoustic Entropy, H (**g**, **h**) in 1354 1°×1° grid squares across North America between 1996 and 2017 and 715 1°×1° grid squares across Europe between 1998 and 2018. Colours indicate the size and direction of trend in each acoustic index (yellow—improving soundscape quality; blue—declining soundscape quality); note that the colour scheme is reversed for AEI, as positive trends are taken to represent a reduction in soundscape quality for this index. Site-level trends are derived from changes in standardised annual values of each acoustic index.

*Europe*: we used annual bird count data from 23 survey schemes across 22 countries collated under the Pan-European Common Bird Monitoring Scheme (PECBMS: https://pecbms.info) from 1998 to 2018. In each scheme, skilled volunteers carry out either line transects, point counts or territory mapping at survey sites during the breeding season and record all birds encountered[60] (Supplementary Table 5); while methods vary between survey schemes, they are consistent within schemes across the time period included here.

Where count data were reported for subspecies, these were aggregated to species level and any records of hybrid species or specifying genus only were removed. The longitude and latitude of each survey site (just the first site of each NA-BBS survey route) were also provided by NA-BBS and PECBMS. Not all sites were surveyed in every year and only sites surveyed at least three times during the defined time period were included in analyses. Note that similar results were found when restricting data to sites surveyed in at least 10 years during the defined period.

**Sound recordings.** Sound files for all species detected on NA-BBS and PECBMS surveys were downloaded from Xeno Canto, an online database of sound recordings of wild birds from around the world (http://www.xeno-canto.org). Specifically, we identified all files longer than 30 s, with associated metadata categorising them as high quality (category "A") and as either "song", "call" or "drumming" types; sound files whose type category including the term "wingbeat", "flap", "begging", "alarm" or "night" types were excluded. Sound files downloaded for NA-BBS species were restricted to those recorded in North America and those from PECBMS to recordings made in Europe. If no sound files met these requirements for a given species, we downloaded all files of shorter duration for that species that met the quality and type criteria and stitched repeats of these together to produce files longer than 30 s. Where more than 50 sound files for a given species met our criteria for inclusion, a random selection of 50 was taken for use in subsequent analyses. We used multiple sound files for each species to capture, where possible, between-individual variation in song and call structure, with the sound file(s) for inclusion in specific soundscapes randomly subsampled from this set. If no sound files for a species were available, the sites where that species was detected were removed from subsequent analyses; this represented <1.5% NA-BBS sites and <3.5% PECBMS sites. Each downloaded sound file was then standardised to ensure consistent sampling rate, duration and volume. Each file was clipped to the first 27.5 s, with the first 2.5 s of this then removed to produce a 25 s recording. These sound files varied in the quantity of vocalisation they contained according to the song and call characteristics of the focal species. Thus, some included 25 s of continuous song while others included just a single, short burst of sound. The sampling rate was set to 44.1 kHz, and each file normalised with a −6 dB gain before being saved as a mono mp3 output.

It is important to recognise that the sound recordings used here are taken in the wild and thus inevitably contain some background noise in addition to vocalisations of the target species, and that this may influence the acoustic properties of the constructed soundscapes to some extent. To minimise this, we selected only Quality "A" recordings and clipped out 25 s from the beginning of each of these for use in soundscape construction, on the assumption that the named focal species will be more dominant in these recordings and that it is most likely to be vocalising towards the beginning of a submitted recording. Furthermore, any background noise is expected to be both random in acoustic structure and randomly distributed across the sound files of species considered here; we see no plausible reason why, for example, the field recordings of increasing or declining species would be more or less likely to contain background noise. Our systematic approach to soundscape construction and our analyses of trends in standardised site-level acoustic metrics also limits the potential of background noise to cause directional bias in the results reported and, if anything, it is expected to have reduced our ability to detect changes in soundscape characteristics.

In total, count data were available for 202,737 sites and 620 species in North America, with a mean ± SE of 15.62 ± 0.6 sound files available per species. For Europe, count data were available for 16,524 sites and 447 species, with 21.05 ± 0.9 sound files per species.

**Soundscape reconstruction.** This is described in detail in the main text.

**Soundscape characteristics.** Four acoustic indices were used to explore changes in the acoustic properties of reconstructed soundscapes. The Acoustic Diversity Index (ADI) uses the Shannon–Wiener index to estimate acoustic diversity, dividing spectrograms into frequency bands and calculating the proportion of each band occupied by sounds above a set amplitude threshold[30]. Higher values represent a more even distribution of sound across frequencies and are associated with increased species richness. The Acoustic Evenness Index (AEI) uses a similar approach, dividing spectrograms into frequency bands but using the Gini coefficient to measure the evenness of sound distribution across them[30]. It is therefore negatively related to ADI, with higher values representing a greater unevenness between frequency bands, suggesting dominance by fewer species. Increases in abundance are expected to have less impact on ADI and AEI than increases in species richness as the songs of individuals from the same species will broadly occupy the same frequency space. The Bioacoustic Index (BI) measures variation in amplitude across a range of frequencies by calculating the dB spectrum across frequencies and quantifying the area under the curve[31]. BI is expected to increase with both increases in abundance and species richness. Total Acoustic Entropy (H) is defined as the product of spatial and temporal entropies and quantifies variation in amplitude across frequency bands and time using Shannon–Wiener index[32]. It increases with both species richness and abundance following a logarithmic model[28,32]. As soundscapes become saturated, the influence of additional species and/or individuals on BI and H is expected to decrease. Default settings were used

**Table 4 Results of GLMMs of the association between site-level trends in acoustic indices and site-level trends in the total number of individuals and species in (i) 202737 NA-BBS sites across North America between 1996 and 2017 and (ii) 16524 PECBMS sites across Europe between 1998 and 2018, and the proportion of 1000 bootstrapped models reporting significant effects for each term ($p < 0.05$).**

|  | Fixed effects | Estimate (SE) | $\chi^2$ | DF | $p$ | Proportion significant ($p < 0.05$) |
|---|---|---|---|---|---|---|
| **(i) North America** | | | | | | |
| (a) ADI | Individuals | 0.109 (0.001) | 310.08 | 1 | <0.001 | 1 |
|  | Species | 0.578 (0.008) | 4955.960 | 1 | <0.001 | 1 |
| (b) AEI | Individuals | −0.053 (0.006) | 72.46 | 1 | <0.001 | 1 |
|  | Species | −0.581 (0.008) | 4964.17 | 1 | <0.001 | 1 |
| (c) BI | Individuals | 0.421 (0.006) | 51.9.51 | 1 | <0.001 | 1 |
|  | Species | 0.615 (0.008) | 6267.72 | 1 | <0.001 | 1 |
|  | Individuals*species | 0.059 (0.147) | 15.975 | 1 | <0.001 | 0.65 |
| (d) H | Individuals | 0.910 (0.005) | 32156.51 | 1 | <0.001 | 1 |
|  | Species | 0.662 (0.007) | 9766.56 | 1 | <0.001 | 1 |
|  | Individuals*species | −0.058 (0.013) | 20.99 | 1 | <0.001 | 0.67 |
| **(ii) Europe** | | | | | | |
| (a) ADI | Individuals | 0.388 (0.029) | 173.51 | 1 | <0.001 | 1 |
|  | Species | 0.657 (0.046) | 206.45 | 1 | <0.001 | 1 |
| (b) AEI | Individuals | −0.398 (0.029) | 182.80 | 1 | <0.001 | 0.98 |
|  | Species | −0.681 (0.046) | 224.51 | 1 | <0.001 | 1 |
|  | Individuals*species | 0.188 (0.086) | 4.80 | 1 | 0.028 | 0.33 |
| (c) BI | Individuals | 0.074 (0.029) | 6.25 | 1 | <0.001 | 0.98 |
|  | Species | 0.947 (0.050) | 424.93 | 1 | <0.001 | 1 |
| (d) H | Individuals | 0.410 (0.029) | 199.76 | 1 | <0.001 | 1 |
|  | Species | 1.095 (0.045) | 590.19 | 1 | <0.001 | 1 |

(a) Acoustic Diversity Index (ADI), (b) Acoustic Evenness Index (AEI), (c) Bioacoustic Index (BI) and (d) Acoustic Entropy (H). Only significant interactions are retained. Site-level trends are derived from changes in standardised annual values of each acoustic index, total number of species, and total number of individuals.

for each acoustic index except BI, where the maximum frequency was set to 22,050 Hz.

We initially generated soundscapes for a series of simulated communities to confirm that the acoustic indices respond as expected when calculated from artificial soundscapes. Firstly, we calculated ADI, AEI, BI and H for soundscapes derived from communities comprising 1 to 10, 20, 30, 40 or 50 individuals of each species in turn. Given the randomised selection of sound files, insertion point and playback volume, we iterated this process 1000 times for each species-abundance combination. Next, we constructed communities containing 2, 3, 4, 5, 10, 20 or 50 species, with 1–10 individuals of each species present, i.e. 70 communities in total. We iterated this process 100 times for each species richness-abundance combination, randomly selecting species for inclusion from the NA-BBS species pool, and a further 100 times, randomly selecting species from the PECBMS species pool. Again, the four acoustic indices were calculated for each soundscape produced.

Annual soundscapes for each NA-BBS and PECBMS site were constructed from each site-year count file and the four acoustic indices were calculated for each. Given the randomised selection of the specific sound file, insertion point, and playback volume used to represent each individual during the construction of each soundscape, this process was iterated five times, with each acoustic index averaged across these five site-year iterations for use in subsequent analyses. For all PECBMS sites and for the first site of each NA-BBS route, the soundscape generated from the fifth iteration was saved as an .mp3 file. All sound file processing and soundscape construction was undertaken using Sound eXchange programme (SoX: http://sox.sourceforge.net/) and acoustic indices were calculated using R packages 'seewave'[32], 'soundecology'[61] and 'tuneR'[62] in R v3.5.1[63].

Finally, we tested the sensitivity of soundscape characteristics to key parameters imposed during construction. While predominately driven by community composition, the acoustic properties of constructed soundscapes could also be influenced by rules that influence the degree of the overlap between individual sound files and their amplitude. First, we generated a community of 10 randomly selected European bird species and specified declines in each species from 10 to five individuals over a 6-year period. For each year, we then constructed four soundscapes and extracted the associated acoustic indices for each. The first soundscape type was built using the methods described above. The second was built by inserting sound files in a 3-min soundscape, to increase the degree of overlap, while the third was built by inserting sound files into a 10-min soundscape to decrease the degree of overlap. Finally, we reverted to a 5-min soundscape but randomly sampled playback volume for each sound file from a half-normal distribution. This increased the relative proportion of distant vocalisations and may be more representative of point count data, where the area surveyed increases with increasing distance; though note this is likely to be offset by reduced detectability at

greater distances. This process was iterated for 1000 randomly sampled communities of 10 species.

### Statistical analyses

*Response of acoustic indices to changes in community structure.* To confirm that acoustic indices respond to changes in species richness and abundance, we fitted General Linear Models (GLMs) to outputs for the simulated single and multi-species communities. In each model, the mean acoustic index across all iterations was fitted as the response variable. For the single-species communities, the log number of individuals was fitted as the explanatory variable and for the multi-species communities, the log number of individuals, log number of species and their interaction were fitted as explanatory variables. Separate models were fitted to the North American and European data and for each acoustic index in turn.

*Site-level changes in acoustic indices.* We standardised each acoustic index within each site (by subtracting the mean site-level measure from the annual value and dividing by the site-level standard deviation[64]) prior to analysis to account for any potential differences in detectability or observer effects between sites, differing sampling protocols across survey schemes, and for initial community structure. In all analyses, separate models were constructed for North American (204,813 sites on 4197 routes spanning 22 years) and European data (16,524 sites spanning 21 years), and for each acoustic index in turn. To explore large-scale temporal trends while accounting for any geographic differences in acoustic characteristics, we fitted Gaussian General Linear Mixed Models (GLMMs) via the R package 'lme4'[65]. Standardised annual site-level values for each acoustic index were fitted as the response variable, with latitude, longitude and year (continuous) as fixed effects. To account for non-independence of soundscapes from the same site, random effects of site and year were included in all models, along with route and state (North America models, Eq. (1a)) or country (Europe models, Eq. (1b)). To assess the importance of fixed effects, we performed a likelihood ratio test by comparing models with and without a particular term, reporting the $\chi^2$ value and associated significance. Spatial autocorrelation of modelled residuals was examined by Moran's I, separately for each year, using the package 'ape'[66]. While significant spatial autocorrelation was found, the sizes of the estimates were negligible (Supplementary Table 6) and therefore this is subsequently ignored. To explicitly explore how temporal trends in the acoustic properties of reconstructed soundscapes varied geographically, we refitted the models described above, including latitude*year and longitude*year interaction terms. To visualise the large-scale annual variation in acoustic properties we refitted these models with year included as a categorial rather than a continuous variable, with predictions from these models providing continent-level annual estimates for each acoustic index (Fig. 3).

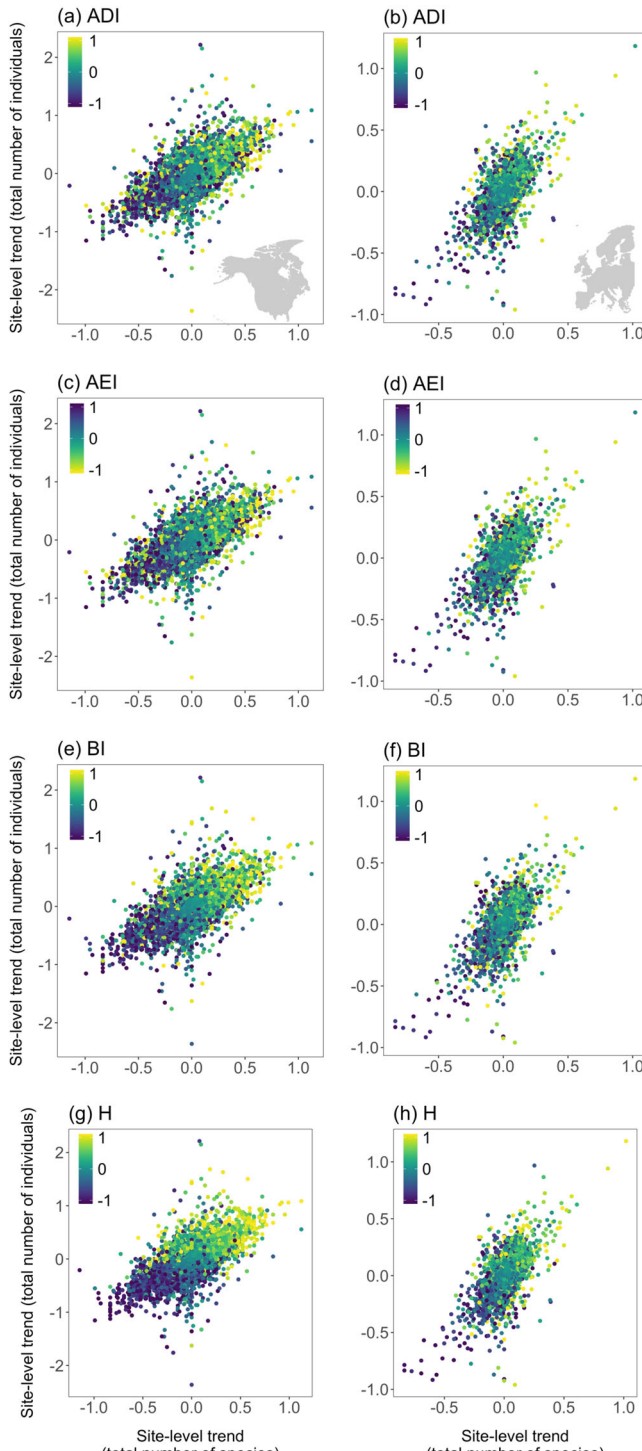

**Fig. 5 Trends in community structure and soundscape characteristics.** The association between site-level in trends in the total number of species and the total number of individuals in 202737 NA-BBS sites across North America (left column) and in 16524 PECBMS sites across Europe (right column). Colours indicate site-level trends in Acoustic Diversity Index, ADI (**a, b**), Acoustic Evenness Index, AEI (**c, d**) Bioacoustic Index, BI (**e, f**) and Acoustic Entropy, H (**g, h**) (yellow—improving soundscape quality; blue— declining soundscape quality; note the colour scheme is reversed for AEI, as positive trends are taken to represent a reduction in soundscape quality for this metric. Site-level trends are derived from changes in standardised annual values of each acoustic index, total number of species, and total number of individuals.

To explore the relationships between site-level trends in each acoustic index, we fitted GLMs with the standardised annual values for each index as the response variable and year (continuous) as the explanatory variable (Eq. (2)). This resulted in an independent estimate of the rate of change in each acoustic index at each site. For all six possible pairwise comparisons between acoustic indices, we used Pearson's correlation coefficients to estimate the magnitude of the association between their site-level trends. All statistical analyses were carried out in R v3.5.1[63].

$$\text{Standardised acoustic index}_{i,t} \sim \beta_0 + \beta_1 \text{Latitude}_i + \beta_2 \text{Longitude}_i + \beta_3 \text{Year}_t$$
$$+ \alpha_{1i} \text{Site}_i + \alpha_{2t} \text{Year}_t + \alpha_{3j} \text{State}_j + \alpha_{4k} \text{Route} + \varepsilon_{i,t}$$
$$(1a)$$

$$\alpha_{1i} \sim N\left(0, \sigma^2_{\alpha_1}\right)$$

$$\alpha_{2t} \sim N\left(0, \sigma^2_{\alpha_2}\right)$$

$$\alpha_{3j} \sim N\left(0, \sigma^2_{\alpha_3}\right)$$

$$\alpha_{4k} \sim N\left(0, \sigma^2_{\alpha_4}\right)$$

$$\varepsilon_{i,t} \sim N(0, \sigma^2_\varepsilon)$$

where $i$ = site, $t$ = year, $j$ = state, $k$ = route

$$\text{Standardised acoustic index}_{i,t} \sim \beta_0 + \beta_1 \text{Latitude}_i + \beta_2 \text{Longitude}_i + \beta_3 \text{Year}_t$$
$$+ \alpha_{1i} \text{Site}_i + \alpha_{2t} \text{Year}_t + \alpha_{3j} \text{Country}_j + \varepsilon_{i,t}$$
$$(1b)$$

$$\alpha_{1i} \sim N\left(0, \sigma^2_{\alpha_1}\right)$$

$$\alpha_{2t} \sim N\left(0, \sigma^2_{\alpha_2}\right)$$

$$\alpha_{3j} \sim N\left(0, \sigma^2_{\alpha_3}\right)$$

$$\varepsilon_{i,t} \sim N(0, \sigma^2_\varepsilon)$$

where $i$ = site, $t$ = year, $j$ = country

$$\text{Standardised acoustic index}_t \sim \beta_0 + \beta_1 \text{Year}_t + \varepsilon_t \qquad (2)$$

$$\varepsilon_t \sim N(0, \sigma^2_\varepsilon)$$

where $t$ = year

To explore large-scale temporal trends in the total number of individuals and species recorded on NA-BBS and PECBMS surveys, we fitted two additional GLMMs. Standardised annual site-level values of the total number of (a) individuals or (b) species were fitted as response variables, with latitude, longitude and year (continuous) as fixed effects. To account for non-independence in community structure from the same site, random effects of site and year were included in all models, along with route and state (North America models) or country (Europe models). Model structures were therefore equivalent to those set out in Eqs. (1a) and (1b), albeit with different dependent variables. We then refitted these models including year as a categorical rather than a continuous variable to visualise the large-scale annual variation, and used predictions from these models to provide continent-level annual estimates for total abundance and species richness (Supplementary Fig. 5).

To explore the site-level relationships between trends in total number of individuals, total number of species and acoustic indices, we first fitted GLMs with either the standardised total number of (a) individuals or (b) species as response variables and year (continuous) as the explanatory variable at each site. These models were therefore equivalent in structure to that described in Eq. (2) and resulted in independent estimates of the rates of change in the total number of individuals and species at each site. We then fitted separate GLMMs for each acoustic index, in each continent, in turn with site-level trend in acoustic index as the response variable and site-level trends in the total number of individuals and the total number of species and their interaction as fixed effects. State was included as a random effect in the North American models and country as a random effect in the European models. To incorporate the error associated with site-level trend estimates we used a bootstrapping procedure in our assessment of the significance of the modelled effects. We generated 1000 new estimates for each variable (site-level trend in: acoustic index, total number of individuals and total number of species) by randomly sampling from a normal distribution with a mean equal to the site-level trend and standard deviation equal to the standard error of the site-level trend. The GLMMs were then fitted over each of the 1000 datasets separately. We present the results of a final model carried out on the original site-level estimates, as well as the proportion of times each fixed

effect included in the final model was significant across the 1000 bootstrapped datasets ($p < 0.05$). Non-significant interaction terms were removed from the models.

**Reporting summary**. Further information on research design is available in the Nature Research Reporting Summary linked to this article.

## Data availability

The North American bird monitoring data are available directly from U.S. Geological Survey (https://www.pwrc.usgs.gov/bbs/) and the European bird monitoring are available, on request, from PECBMS (https://pecbms.info/). Sound recordings were downloaded from Xeno Canto (http://www.xeno-canto.org). Acoustic indices for soundscapes constructed from simulated communities, site-level acoustic index data for reconstructed soundscapes for NA-BBS and PECBMS sites, and source data for all figures are available from the Open Science Framework under accession code: https://osf.io/jyuxk/ (ref. [67]).

## Code availability

R code for soundscape construction, extraction of acoustic indices, statistical analyses and figure construction are available from the Open Science Framework under accession code: https://osf.io/jyuxk/ (ref. [67]).

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

## Acknowledgements

We thank the thousands of volunteer citizen-scientists who contributed to the long-term bird-monitoring programmes in North America and Europe, the institutions that manage these programmes, and those funding these activities. The Norwegian Environment Agency finances the Norwegian common breeding bird monitoring and the Swedish Bird Survey is supported by the Swedish Environmental Protection Agency and carried out in collaboration with all 21 regional county boards. It acts within the framework of the strategic research environment Biodiversity and Ecosystem Services in a Changing Climate (BECC). J.R. was supported by Charles University in Prague (project no. PRIMUS/17/SCI/16). We are also grateful to the many sound recorders that have submitted files to the Xeno Canto collection (www.xeno-canto.org). This work was supported by Natural Environment Research Council grant NE/T007/354/1. We thank L. Spurgin for advice on analysis; and J. Gill and J. Sauer for helpful discussions and comments on the manuscript. Soundscape reconstruction was carried out on the High-Performance Computing Cluster supported by the Research and Specialist Computing Support Service (RSCSS) at the University of East Anglia.

## Author contributions

S.J.B. conceived the study and constructed the soundscapes. S.J.B. drafted the manuscript with significant contributions from C.A.M. C.A.M. performed the data analyses and prepared the figures. A.A., Z.B., L.B., T.C., P.C., V.E., D.P.E., A.G., S.H., F.J., J.A.K., J.K., A.K., P.K., A.L., Å.L., C.M., D.G.N., I.J.Ø., J-Y.P., J.R., T.S., B.S.S., N.T., S.T., C.A.M.v.T. and P.V. contributed to collection and collation of PECBMS data. All authors reviewed and approved the final manuscript.

## Competing interests

The authors declare no competing interests.
