## [Peer Review File · Nature Communications]

Peer Review Comments, initial response to Manuscript:

Reviewer #1 (Remarks to the Author):

This is a fascinating study in which the authors examine the relationship between biodiversity loss and soundscape quality. The authors hypothesized that increased biodiversity loss will result in less complex soundscapes, which could potentially affect human health and well-being through a “diluted experience” with nature. To test this hypothesis, the authors use an impressively large dataset on birds—from bird-monitoring programs in North America and Europe—and the xeno-canto database of sound recordings to reconstruct historical soundscapes, spanning a period of 25 years (1996/1998 to 2017/2018). The authors quantified the quality of the soundscapes using four different acoustic indices. They found that soundscape quality has indeed decreased over time but the results varied spatially, with some areas in North America and Europe showing the opposite patterns. Overall, the study is well-written and it could make a very interesting contribution to the literature. That said, there are a few issues that the authors may want to consider:

1. I was somewhat surprised to see that there wasn't a more detailed analysis of the changes in the bird communities in the sites used in the study. Although I understand that the study's main objective was to evaluate the soundscapes and their change over time (by measuring acoustic diversity and intensity), a key assumption of the study is that this acoustic change is caused by changes in the bird communities. Wouldn't be therefore useful to match the acoustic analysis to an analysis that explores the corresponding changes in bird species richness and composition over time? These factors can influence significantly the acoustic indices. Currently, there little information in the manuscript about how the bird communities might have changed over time and to what extent, and hence it is impossible to know what is exactly is causing the patterns reported in the study.
2. Moreover, Figures 5 and S2, for example, show that there is considerable spatial variation in the reported patterns. In many places, the soundscape quality has actually increased over time. This is a highly interesting pattern that somewhat “contradicts” the main message of the study. I wondered whether such an analysis, i.e., of the changes in the bird communities, could have helped understand better these patterns. Currently, the potential explanations provided by the authors (e.g., lines 177-185) are somewhat speculative and do not provide much evidence.
3. On a similar note, could these spatial differences be due to methodological issues? For example, as the authors mention in their manuscript, not all sites were sampled every year. Some (unclear how many) were sample for only three years out of the 25 years. Is it possible that the results would be more spatially consistent if the authors had used a stricter cut-off point (e.g., 10 vs. 3)?
4. The manuscript can be strengthened by adding some more details about the methods used, which would allow readers to evaluate better the analytical framework. The analysis involves many different regression models but it is not always clear what was done and why. To give a simple example, in Figure 3, the authors mention that the results are based on a model in which “year” was used as a categorical variable. However, this particular model is not described/explained in the methods. It is, therefore, not obvious why it was necessary to run a nearly-identical model twice, i.e., once with the “year” as a continuous variable and once with the “year” as a categorical variable. I would suggest that the authors expand their description of the models (lines 594-606) to explain in more detail: (a) what was each model testing exactly, (b) which variables were used in each model (ideally by providing the full equation each time), and (c) what was the actual sample size used.
5. The same applies to the methods concerning the reconstruction of the soundscapes. It seems to

me that the authors have reconstructed a series of different soundscapes that were necessary for the different parts of the analysis. For example, there are the soundscapes used to test how well the four acoustic indices respond to artificially made soundscapes (e.g., the 2,3,4,5,10,20,50 files), and then there are the actual “annual soundscapes”, which is not exactly clear how they were created (please see a related comment below).

Other points:

Lines 98-102: As the authors correctly point out, the relationship between the acoustic indices and bird abundance and diversity is often non-linear. Presumably, this was the reason the number of individuals was log-transformed. Yet, this was not done for “species richness,” and, therefore, a linear relationship was assumed despite the wide range of species (e.g., see Table 2).

Line 105-114: I find the description here somewhat vague. Were the “individuals of all other species” added to the same sound file or to a different file? I would assume that they were all added to the same file to recreate a soundscape that reflected the species at the particular site/year. But the captions of Table 1 and Figure 3, suggest that the analysis is based on single-species reconstructed soundscapes. If that is the case, how representative can single-species soundscapes be of the real soundscapes at each site/year? Maybe I am reading the caption of Figure 3 incorrectly, particularly the parts that say “lines show the predicted trends from GLMMs (Table 1)” and “GLMMs with the identical structure as those in Table 1”.

Line 109: Was the duration of all the files 5 minutes? Even those with 50 species? I would think that for the results of the acoustic indices to be comparable, the total duration of the reconstructed files should be the same.

Line 515: What about other background noises often found in xeno-canto files, which may have influenced the results (especially for species with few recordings in the database)?

Line 568: Perhaps use “2,3,4,5,10,20,50”, as in the caption of Table 3, so that it is clear you had seven categories.

Line 577-578: What was the purpose of saving this last “mp3” file? Please clarify.

Line 603-604: Does this mean that the same variable, i.e., “year,” was included as a random and as a fixed effect? Why was this necessary?

Line 608-609: It is unclear why the spatial autocorrelation was “examined separately for each year.” The reported temporal patterns are based on models that included all years (e.g., in Table 4); therefore, why not examine the spatial autocorrelation of those models instead (which are the models on which the main findings are based). Also, shouldn’t the authors test for other issues that may have violated the assumption of independence, e.g., temporal autocorrelation.

Reviewer #2 (Remarks to the Author):

Overview

The authors reconstructed soundscapes in Europe and North America based on bird survey data to quantify how the human experience of nature may have changed over 25 years. This is a very creative way of exploring an important topic. The authors found that acoustic complexity and diversity have generally decreased, but that change was heterogeneous in space and time. This has global scope and – if the analyses are sound – potentially significant implications for changes to the human experience.

Major Comments

The paper would benefit from an articulation of a testable hypothesis with quantitative predictions. Based on what is presented, the authors assume that acoustic indices represent the human experience of soundscapes and confirm a positive association between the indices and biodiversity. Thus, their implicit hypothesis seems to be that if bird communities change through time, human auditory experience will change too. This idea is present throughout, but I think a nice direct statement would be good.

One major concern is that the authors do not mention taking any steps to ensure that each species' recording contained only the species of interest. Although some (many?) xeno-canto recordings are produced with parabolic microphones that should exclude non-target sounds, there are plenty of examples in which multiple species are singing. Longer recordings, which the authors used, are more likely to be susceptible to this problem. The distinct possibility that multiple species are present in a recording, and thus that the reconstructed soundscapes are biased depending on their species composition, needs to be addressed in some way. Maybe (hopefully!) an explanation of quality control measures that goes beyond XC's "A=loud and clear" would help, but this does seem like a potentially serious issue.

Another major concern relates to playback volume. If playback volume was randomly allocated (109-110), that suggests that volume was drawn (randomly) from a uniform distribution. If so, that would mean a bird is equally likely to be "close" or "far" from the hypothetical observer (ie, loud or quiet). Yet if we assume that birds are uniformly distributed in space, the number of birds encountered should increase exponentially with distance (up to some perceptual limit) because exponentially more area is included in our search. Drawing from a uniform distribution means that the density of birds is highest at the point of observation because no more birds are observed despite a greater area. Thus, playback volume should be randomly drawn from an exponential, log-normal, beta, or other, similar, distribution. I expect this could make the soundscape indices unnaturally sensitive to changes in abundance. This issue could be particularly problematic for common, abundant, and vocally active species like the American Robin or Eurasian Blackbird: multiple individuals are likely to be present at any given site and – under the uniform distribution scenario – their songs could artificially dominate the soundscape. I'm concerned that – if I've understood what the authors have done – this feature of the reconstruction is unrealistic and can introduce substantial bias into the analyses.

Another quantitative remark is that US states and European countries do not seem like suitable random effects when modeling the acoustic indices (605-606). This assumes that the soundscapes across those areas have one mean value which has been repeatedly sampled, yet many US states and European countries are sufficiently large that this assumption is not accurate. Consider the soundscapes in the Italian Alps vs Sicily, or in the temperate rainforests of northern California compared to the desert of southern California – why would we expect those soundscapes to not be

independent? I wonder if the authors could actually improve their explanatory power by removing these random effects; replacing them with an ecological units such as biomes/ecoregions – rather than arbitrarily defined political constructs – might be a better option.

I think the paragraph about acoustic indices (118-132) would benefit from reorganization: I think it would make more sense to talk about the correlation between the indices and species richness in the reconstructed soundscapes before talking about the connection between indices and real-world data. This would flow better from the preceding paragraph in which the reconstructed soundscapes are discussed, so the conceptual transition is smoother.

In the results (158-176), there is a lot of description of change in acoustic indices but I'm left wondering what it all means in biological or experiential terms. Some translation to that perspective would be nice.

Minor Comments

Title: this is a pretty trivial point, but given that the NA-BBS data are collected in June, you've really reconstructed summer soundscapes not spring ones. Given the parallel the authors are making with Rachel Carson's famous book, leaving the title as-is is probably fine.

61: "data-driven" feels more like a corporate catch-phrase than something meaningful.

65-66: "quality" is subjective and not informative, but "acoustic diversity and intensity" are useful. Cut "quality".

74: this is more of an aside, but do you think that framing the "extinction of experience" as a "human-nature disconnect" actually contributes to the problem by reinforcing a conceptualization of humans as fundamentally separate from nature?

75-76: the detrimental affects of school/work closures seems to be a social issue. It seems like the authors are stretching to connect a hot issue (covid-19) to their work.

77: typo? delete "necessarily"

81: the authors answer the "extent" question but only provide some speculative "implications." For this reason, I would remove implications from this sentence.

87, 95: could delete "Indeed" without affecting the meaning of the sentence

94: consider replacing "impacting" with "affecting"

108: why specify that it was the first species when there was nothing special about it being first? Just say "for each species"

124: delete comma after "intensity", add comma after "therefore"

125: "quality" and "value" are subjective and uninformative.

140: "quality" is finally defined here, but rather poorly. The authors write "...changes in soundscape quality, defined as a reduction in acoustic diversity and/or intensity...". This suggests that quality = low diversity/intensity, yet this is the opposite of what the authors seem to mean. I think "quality" is quantified by acoustic diversity/intensity, with high quality = high diversity/intensity and low quality = low diversity/intensity.

143-145: Is this statement about soundscape changes bases on the authors' findings, or on published research? Either a reference is missing, or the authors should combine this sentence with the following one. As written, it gives the impression that the authors have reached this conclusion before their analyses were conducted. Instead, this statement could be framed as a hypothesis which the authors set out to test.

165: "lesser"

166 and other places: "strongly" refers to force but the word is modifying a quantity, so something like "substantially" would be more appropriate

170-174: “fast” and “faster” across several sentences and it gets confusing

183: “regional, biome, and local” should be ordered by scale (one way or the other), and it isn’t really clear that this is the case

183-4: typo? What is “fundamental species richness and abundance?”

189-190: “will only exacerbate this process”

206-213: the recommendation that soundscape data be collected should be refined. Many hundreds of TB of soundscape data exist and are being generated each year by acoustic monitoring projects, but that data tend to be collected in relatively intact ecosystems where humans are not routinely experiencing the soundscape. My point is that this suggestion seems too general and not particularly novel. What exactly would the authors want to see from a “soundscape monitoring scheme”? Recording units across urban to rural to wilderness gradients?

215: I’m in favor of the Oxford comma

216: sound is more defining than visual? I disagree

216-219: this sentence needs work. Clarify that the authors are reporting these declines, clarify that the declines are based on reconstructions (not measured declines), consider removing “concerning”

226: delete “resultant”

226: why is “dilution of experience” in quotes throughout the manuscript?

228-23: the authors state that “conservation policy and action need” to protect soundscapes. This raises a question about what auditory interactions with birds represent. Do people care about the sounds themselves or what they understand the sounds to represent? A conservation policy based on soundscapes alone could be to simply play recordings of complex soundscapes (e.g., reference 15). Would the authors find this solution sufficient?

All Figures: The text (axes, legends, etc.) is very small and hard to read

451: active voice

562: inserting a dependent clause after “that” makes the sentence hard to follow.

563: it seems like a comma is missing after “expected”.

Figure S1: the color scheme is not easy to discern with such small maps; why not use the spectrum used for Fig 5?

Reviewer #3 (Remarks to the Author):

This paper is extremely creative and novel; I love the central idea and I think it adds a compelling new dimension to how we think about global change and the impact it is having on biodiversity, and also the human experience of nature. While there was much to like about the paper, I also thought there were a number of areas in which it could be improved.

1. I felt the topping and tailing with an extinction of experience narrative was a little incongruous, and included a number of rather speculative statements, e.g. the implication that more diverse bird song leads to greater connection to nature and therefore enhanced conservation action (for counter-examples and nuance, see e.g. Pinder et al. 2020; Oh et al 2021). [regarding the health and wellbeing outcomes of nature experiences, I agree with the authors that the evidence is clear]. As well as being rather tenuous, this narrative was distracting and is not really needed to weave the story of a degradation of natural soundscapes. In any case, the paper doesn’t actually analyse anything to do with the human experience, or require recourse to any of the more speculative links between those experiences and conservation concern / action. Consider dramatically downsizing the

opening and closing remarks, or reframing the first and last paragraphs to focus more simply and directly on the issue being studied.

2. Much valuable real estate in the main paper is taken up with the results of simulations. I think much of this could be moved to supplementary information, and replaced with a deeper dive into what is driving the empirical long term changes in the acoustic indices. I really wanted to know what this soundscape-reconstruction approach is telling us that we can't get from analysing the underlying survey data – this seems to be important in justifying this approach and contributing something truly new to the literature. For example, are there non-linear dynamics, such that the soundscape suddenly falls apart as tipping points are breached, what is the relative role of species richness and abundance in explaining how rapidly the soundscape deteriorated at a site. How does composition versus species richness matter? Are increasing species (e.g. urban-adapted, invasives), contributing a particular sort of acoustic signal that is driving some of the changes.

3. There is no validation of the method of constructing and analysing the sound segments. Maybe I'm missing something, but I would have thought that a field validation would be rather straightforward – doing some point counts at the same time as soundscape recordings are being made, then constructing the file segments via xeno canto etc as per the methods, then comparing that statistically with the soundscape recordings. As well as comparing the indices, one could compare the number of calling events by different species etc. This might help underpin decisions about how to optimally translate the results of a survey into a sound file, and showing that the method works in a number of different settings, e.g. different habitats. This sort of underpinning validation would also give us some confidence that the indices arising from the constructed sound files actually represent a decent approximation of the soundscape at the time of the original survey.

4. The protocols for the NA-BBS and PECBMS surveys are exceedingly different, with the former being a 3-min point count of birds within 400m and the latter being a mix of "line transects, point counts or territory mapping" including all birds encountered. The authors make the brief argument that because these are internally consistent there is no problem. I'm not sure about that for two reasons. First, there could be substantial geographic heterogeneity in the types of methods employed in various parts of Europe, or temporal trends in the sorts of methods applied across the time series, leading to systematic differences in the type of data obtained. How does one turn "territory mapping" into a simple species list used in the analysis? Second, and given that the relationship between number of indivs / species and soundscape quality is nonlinear, there might be non-linear effects on the results, because the Euro soundscapes will be far denser than North America given the more comprehensive species lists collected over a longer period. The vast differences in methods at least needs to be considered a bit more comprehensively.

MINOR COMMENTS

5. There is a one-size-fits-all approach to constructing the sound files (25s soundbites regardless of the type of species involved – I appreciate that includes some natural spacing of vocalisations within that segment), and I wonder instead if these could be based on estimates of calling rates, bout durations etc from the literature? The avifaunas of Europe and N America are well served with detailed handbooks containing this sort of information.

6. Recordings of restricted species are not directly available in xeno canto, and the fact that sites containing one or more species without a recording were removed could mean that, e.g. wilderness

areas are underrepresented? This could be a systematic problem if wilderness areas are less likely to have undergone bird declines. Could we see how many (and maybe also where) the removed sites were? Did you consider using an alternative source (e.g. eBird / Macauley library collection, commercial recordings) as an alternative to fill some of these gaps?

7. Line 207: For interest, albeit at the risk of self-promotion (I have signed this review), I'm keen to point the authors to the Australian Acoustic Observatory (<https://acousticobservatory.org>) – a methods paper describing this network has just been accepted in MEE (Roe et al. in press).

Oh RRY, Fielding KS, Nghiem LTP, Chang C-C, Carrasco RTL & Fuller RA (2021) Connection to nature is predicted by family values, social norms and personal experiences of nature. *Global Ecology and Conservation*, 28, e01632.

Pinder J, Fielding KS & Fuller RA (2020) Conservation concern among Australian undergraduates is associated with childhood socio-cultural experiences. *People and Nature*, 2, 1158-1171.

Roe P, Eichinski P, Fuller RA, McDonald P, Schwarzkopf L, Towsey M, Truskinger A, Tucker W & Watson D (in press, accepted 29 Apr 2021) The Australian Acoustic Observatory. *Methods in Ecology and Evolution*.

Richard Fuller, University of Queensland

Reviewer #1:

This is a fascinating study in which the authors examine the relationship between biodiversity loss and soundscape quality.. Overall, the study is well-written and it could make a very interesting contribution to the literature.

That said, there are a few issues that the authors may want to consider:

1. I was somewhat surprised to see that there wasn't a more detailed analysis of the changes in the bird communities in the sites used in the study. Although I understand that the study's main objective was to evaluate the soundscapes and their change over time (by measuring acoustic diversity and intensity), a key assumption of the study is that this acoustic change is caused by changes in the bird communities. Wouldn't be therefore useful to match the acoustic analysis to an analysis that explores the corresponding changes in bird species richness and composition over time? These factors can influence significantly the acoustic indices. Currently, there little information in the manuscript about how the bird communities might have changed over time and to what extent, and hence it is impossible to know what is exactly is causing the patterns reported in the study.

As recognised by the reviewer, our paper is predominantly focused on exploring continent-wide changes in soundscape characteristics for the first time, so we had restricted our exploration of the link between community change and the response of acoustic metrics to simulated communities in the original submission (Figs 1 and 2 and associated analyses). In this revision we have substantially extended our analyses of this relationship, using NA-BBS and PECBMS monitoring data to explore the change in each acoustic metric at a site in response to changes in both species richness and the total abundance of individuals at that site (Fig. 5; Table 4). We also report general patterns of site-level changes in species richness and total abundance across both North America and Europe over the past 25 years (Supplementary Fig 4; Supplementary Table 3). These additional analyses match the scale and resolution of our analyses of soundscape characteristics and provide significant insight into the fundamental mechanisms influencing changes in soundscape characteristics. As anticipated, our new analyses reveal strong positive relationships between changes in community composition and acoustic metrics but also show substantial context-dependency. We believe these findings add greatly to the paper and thank the reviewer for suggesting we include these additional analyses.

2. Moreover, Figures 5 and S2, for example, show that there is considerable spatial variation in the reported patterns. In many places, the soundscape quality has actually increased over time. This is a highly interesting pattern that somewhat "contradicts" the main message of the study. I wondered whether such an analysis, i.e., of the changes in the bird communities, could have helped understand better these patterns. Currently, the potential explanations provided by the authors (e.g., lines 177-185) are somewhat speculative and do not provide much evidence.

As discussed above, the additional analyses we now include – reporting both continent-wide trends in site-level species richness and total abundance, and significant positive relationships between site-level trends in acoustic indices and site-level trends in species richness and abundance – fully address this point. The main message is that there have

been pervasive, chronic declines in soundscape quality across North America and Europe over the past 25 years but that this is underpinned both by large-scale geographical patterns and by substantial site-level variation. Our new analyses demonstrate that these changes mirror changes in species richness and abundance, as reported here (Fig 5; Supplementary Fig. 4) and elsewhere (e.g. Rosenberg *et al.* (2019) *Science* 366: 120-124; Morrison *et al.* (2021) *Proc Roy Soc B* 288: 20202955).

3. On a similar note, could these spatial differences be due to methodological issues? For example, as the authors mention in their manuscript, not all sites were sampled every year. Some (unclear how many) were sample for only three years out of the 25 years. Is it possible that the results would be more spatially consistent if the authors had used a stricter cut-off point (e.g., 10 vs. 3)?

We included *Year* as a random effect in our models (see response to point 4 below) to account for differences in which sites were surveyed in each year, so the spatial patterns reported are not being driven by variation in sampling effort across sites. As suggested, we have now repeated our analyses of temporal and spatial trends in acoustic indices but this time restricting the sites included to those surveyed in at least 10 years over the same time period. This reduced the number of survey sites from: US-BBS: 202737 to 148046; PECBMS: 16524 to 7541. These analyses reveal very similar patterns to those currently reported:

Results of GLMMs of the variation in a) Acoustic Diversity Index (ADI), (b) Acoustic Evenness Index (AEI), (c) Bioacoustic Index (BI) and (d) Acoustic Entropy (H) in 148046 BBS sites across North America surveyed in at least 10 years between 1996 and 2017. For comparison, this is equivalent to Supplementary Table 1.

	Fixed effects	Estimate (SE)	χ^2	DF	p
(a) ADI	Latitude	0.00059 (0.00022)	0.05	1	0.8213
	Longitude	-0.00047 (0.00009)	0.06	1	0.8064
	Year	0.00342 (0.00103)	11.28	1	0.0008
	Latitude*year	-0.00005(0.00002)	9.38	1	0.0022
	Longitude*year	0.00004 (0.00001)	40.11	1	<0.001
(b) AEI	Latitude	-0.00095 (0.00022)	0.06	1	0.7993
	Longitude	0.00056 (0.00009)	0.05	1	0.8291
	Year	-0.00601 (0.00095)	9.19	1	0.0024
	Latitude*year	0.00008 (0.00002)	24.14	1	<0.001
	Longitude*year	-0.00005 (0.00001)	56.03	1	<0.001
(c) BI	Latitude	0.00170 (0.00022)	0.00	1	0.9960
	Longitude	0.00040 (0.00009)	0.14	1	0.7036
	Year	-0.00005 (0.00119)	9.03	1	0.0027
	Latitude*year	-0.00014 (0.00002)	73.28	1	<0.001
	Longitude*year	-0.00003 (0.00001)	25.59	1	<0.001
(d) H	Latitude	-0.00110 (0.00022)	0.07	1	0.7982
	Longitude	0.00139 (0.00008)	0.22	1	0.6369
	Year	-0.02194 (0.00204)	14.57	1	<0.001
	Latitude*year	0.00009 (0.00002)	32.04	1	<0.001
	Longitude*year	-0.00011 (0.00001)	325.85	1	<0.001

Results of GLMMs of the variation in a) Acoustic Diversity Index (ADI), (b) Acoustic Evenness Index (AEI), (c) Bioacoustic Index (BI) and (d) Acoustic Entropy (H) in 7481 sites across Europe surveyed in at least 10 years between 1998 and 2018. Only significant interaction effects are retained. For comparison, this is equivalent to Supplementary Table 2.

	Fixed effects	Estimate (SE)	χ^2	DF	p
(a) ADI	Latitude	0.0069 (0.0018)	0.01	1	0.987
	Longitude	-0.0117 (-12.369)	0.76	1	0.385
	Year	0.0172 (0.0081)	8.12	1	0.004
	Latitude*year	-0.0006 (0.0001)	17.57	1	<0.001
	Longitude*year	0.0010 (0.0001)	190.35	1	<0.001
(b) AEI	Latitude	-0.0067 (0.0018)	0.05	1	0.8163
	Longitude	0.0103 (0.0009)	0.65	1	0.4187
	Year	-0.0201 (0.0080)	4.61	1	0.0318
	Latitude*year	0.0006 (0.0001)	17.46	1	<0.001
	Longitude*year	-0.0009 (0.0001)	147.58	1	<0.001
(c) BI	Latitude	-0.0003 (0.0007)	0.15	1	0.7001
	Longitude	0.0002 (0.0004)	0.31	1	0.5749
	Year	-0.0078 (0.0026)	8.52	1	0.0035
(d) H	Latitude	0.0067 (0.0018)	0.19	1	0.6657
	Longitude	-0.0058 (0.0009)	0.60	1	0.4386
	Year	0.0166 (0.0081)	11.89	1	<0.001
	Latitude*year	-0.0006 (0.0001)	17.97	1	<0.001
	Longitude*year	0.0005 (0.0001)	49.12	1	<0.001

Predicted temporal trends in Acoustic Diversity Index, ADI (a,b), Acoustic Evenness Index, AEI (c,d), Bioacoustic Index, BI (e,f) and Acoustic Entropy, H (g,h) in sites across North America surveyed in at least 10 years between 1996 and 2017. For comparison, this is equivalent to Supplementary Figure 2.

Predicted temporal trends in Acoustic Diversity Index, ADI (a,b), Acoustic Evenness Index, AEI (c,d), Bioacoustic Index, BI (e,f) and Acoustic Entropy, H (g,h) in sites across Europe surveyed in at least 10 years between 1998 and 2018. For comparison, this is equivalent to Supplementary Figure 3.

We have added a sentence to the manuscript stating that the results presented are equivalent if restricting data to sites surveyed in at least 10 years (Ln 629).

4. The manuscript can be strengthened by adding some more details about the methods used, which would allow readers to evaluate better the analytical framework. The analysis involves many different regression models but it is not always clear what was done and why. To give a simple example, in Figure 3, the authors mention that the results are based on a model in which “year” was used as a categorical variable. However, this particular model is not described/explained in the methods. It is, therefore, not obvious why it was necessary to run a nearly-identical model twice, i.e., once with the “year” as a continuous variable and once with the “year” as a categorical variable. I would suggest that the authors expand their description of the models (lines 594-606) to explain in more detail: (a) what was each model testing exactly, (b) which variables were used in each model (ideally by providing the full equation each time), and (c) what was the actual sample size used.

We refitted our models with *year* as a categorical rather than a continuous variable to provide continent-level annual estimates for each acoustic metric, and to visualise the large-scale annual variation in acoustic characteristics. We have made significant revisions to the text in this section (ln 731) to clarify and justify our modelling framework, including the addition of model equations (ln 765), and have added details of sample sizes (ln 731).

5. The same applies to the methods concerning the reconstruction of the soundscapes. It seems to me that the authors have reconstructed a series of different soundscapes that were necessary for the different parts of the analysis. For example, there are the soundscapes used to test how well the four acoustic indices respond to artificially made soundscapes (e.g., the 2,3,4,5,10,20,50 files), and then there are the actual “annual soundscapes”, which is not exactly clear how they were created (please see a related comment below).

We constructed soundscapes from simulated communities to examine how the acoustic metrics responded to changes in species richness and/or abundance. The results of these analyses are presented in Figs 1&2 and Tables 1&2. We then constructed soundscapes from the annual, site-level species counts reported by NA-BBS and PECBMS to examine how the acoustic metrics have changed over time across North America and Europe. The results of these analyses are presented in Figs 3&4 and Table 3. Both steps used the same soundscape construction protocol, as described in Lines 120-130. For the simulated communities, we used what was effectively a single site-year count file whilst for the US-BBS and PECBMS data we had separate files for every year in which a site was surveyed. We have made numerous changes to the text in response to other comments which clarify this even further.

Other points:

Lines 98-102: As the authors correctly point out, the relationship between the acoustic indices and bird abundance and diversity is often non-linear. Presumably, this was the reason the number of individuals was log-transformed. Yet, this was not done for “species richness,” and, therefore, a linear relationship was assumed despite the wide range of species (e.g., see Table 2).

We have re-run the analyses of acoustic indices of soundscapes constructed for simulated communities using log-transformed species richness as suggested. The updated analyses, which reveal the same pattern as previously presented, are now reported in Table 2.

Line 105-114: I find the description here somewhat vague. Were the “individuals of all other species” added to the same sound file or to a different file? I would assume that they were all added to the same file to recreate a soundscape that reflected the species at the particular site/year. But the captions of Table 1 and Figure 3, suggest that the analysis is based on single-species reconstructed soundscapes. If that is the case, how representative can single-species soundscapes be of the real soundscapes at each site/year? Maybe I am reading the caption of Figure 3 incorrectly, particularly the parts that say “lines show the predicted trends from GLMMs (Table 1)” and “GLMMs with the identical structure as those in Table 1”.

Firstly, we apologise for an error in the legend of Figure 3 – this incorrectly referenced Table 1 rather than Table 3. We appreciate that this would have caused the confusion expressed here and hope that correcting this addresses that. We have also added some additional text to our description of the process to explain it further (Ln 120). For clarity, a single soundscape was constructed from each site-year count file, with sound files from all contributing species overlaid to produce a composite soundscape. The number of species included, and number of individuals of each of those species, depended on the structure of either the simulated community or what was counted at a given site in a given year. Thus, for the analyses presented in Table 1 and Fig 1 only one species contributed to each soundscape as this component explores the response of acoustic metrics to changes in the number of individuals contributing to a soundscape when only a single species is present. For the analyses presented in Table 2 and Fig 2, both the number of individuals (1-10) and number of species (2,3,4,5,10,20,50) contributing to the soundscape was systematically adjusted by manipulating the simulated communities. For the analyses based on the NA-BBS and PECBMS count data, the number of species and individuals contributing to the soundscape was determined by what was counted at that site in that year.

Line 109: Was the duration of all the files 5 minutes? Even those with 50 species? I would think that for the results of the acoustic indices to be comparable, the total duration of the reconstructed files should be the same.

Yes, this is the correct interpretation of the approach.

Line 515: What about other background noises often found in xeno-canto files, which may have influenced the results (especially for species with few recordings in the database)?

Over 19000 sound files met our criteria for inclusion so it was not possible to “clean” each file prior to inclusion. We targeted high quality recordings (Quality=A) on the assumption that the focal species would be the dominant sound in these files but accept that there is some background noise in them – this is an inevitable consequence of citizen science sound recording in the wild, but these analyses would not have been possible without using such recordings. However, we do not believe background noise imposes any directional bias on our findings – if anything, it reduces our ability to detect changes in soundscape structure as it literally and statistically adds “noise” to the data. For example, there is no reason why increasing/declining species would be more or less likely to have background noise in their sound files. Similarly, background noise is no more or less likely to be present in the sound files of species with fewer recordings.

Line 568: Perhaps use “2,3,4,5,10,20,50”, as in the caption of Table 3, so that it is clear you had seven categories.

Changed as suggested

Line 577-578: What was the purpose of saving this last “mp3” file? Please clarify.

We did this so that we now have audio files of constructed soundscapes for every NA-BBS and PECBMS site in every year that it was surveyed. These can and do provide a fantastic resource for dissemination, public engagement and further investigation (an example can be found here: <https://tinyurl.com/jdcucd34>).

Line 603-604: Does this mean that the same variable, i.e., “year,” was included as a random and as a fixed effect? Why was this necessary?

Yes, *Year* was included as a continuous fixed effect to quantify the rate of change in acoustic metrics and as a random effect to account for the differences in sampling locations between years (not all sites were sampled in all years).

Line 608-609: It is unclear why the spatial autocorrelation was “examined separately for each year.” The reported temporal patterns are based on models that included all years (e.g., in Table 4); therefore, why not examine the spatial autocorrelation of those models instead (which are the models on which the main findings are based). Also, shouldn’t the authors test for other issues that may have violated the assumption of independence, e.g., temporal autocorrelation.

This is a misunderstanding. We have examined the spatial autocorrelation of the models presented in Table 3 (previously Table 4). However we were not able to run this analysis across all years at the same time due to the associated computational demands. We therefore ran, and present, the Moran’s I analysis for each year separately. As suggested by the reviewer we have now also tested for temporal autocorrelation using the acf function in R and find no evidence of temporal autocorrelation in any of our models.

Reviewer #2:

Overview

This is a very creative way of exploring an important topic. The authors found that acoustic complexity and diversity have generally decreased, but that change was heterogeneous in space and time. This has global scope and – if the analyses are sound – potentially significant implications for changes to the human experience.

Major Comments

The paper would benefit from an articulation of a testable hypothesis with quantitative predictions. Based on what is presented, the authors assume that acoustic indices represent the human experience of soundscapes and confirm a positive association between the indices and biodiversity. Thus, their implicit hypothesis seems to be that if bird communities change through time, human auditory experience will change too. This idea is present throughout, but I think a nice direct statement would be good.

Yes, this is our central thesis. We have added a direct statement to this effect (Ln 159), along with clarifying text elsewhere (e.g. Ln 138). The link between changes in bird

community structure and acoustic characteristics is also much more strongly demonstrated with the addition of new analyses.

On major concern is that the authors do not mention taking any steps to ensure that each species' recording contained only the species of interest. Although some (many?) xeno-canto recordings are produced with parabolic microphones that should exclude non-target sounds, there are plenty of examples in which multiple species are singing. Longer recordings, which the authors used, are more likely to be susceptible to this problem. The distinct possibility that multiple species are present in a recording, and thus that the reconstructed soundscapes are biased depending on their species composition, needs to be addressed in some way. Maybe (hopefully!) an explanation of quality control measures that goes beyond XC's "A=loud and clear" would help, but this does seem like a potentially serious issue.

As discussed above in our response to Reviewer 1 on a similar point, we acknowledge that there is background noise in many of the recordings but can see no mechanism by which this would systematically bias our results. We selected Quality "A" recordings and clipped out 25s from the beginning of each of these on the assumption that the named focal species will be dominant and is most likely to be vocalising towards the beginning of a submitted recording. The presence and nature of any background noise is expected to be random across both the sound files of the same species and the sound files of different species. Furthermore, all our analyses are undertaken on standardised, site-level data to account for, amongst other things, differences in initial species composition.

Another major concern relates to playback volume. If playback volume was randomly allocated (109-110), that suggests that volume was drawn (randomly) from a uniform distribution. If so, that would mean a bird is equally likely to be "close" or "far" from the hypothetical observer (ie, loud or quiet). Yet if we assume that that birds are uniformly distributed in space, the number of birds encountered should increase exponentially with distance (up to some perceptual limit) because exponentially more area is included in our search. Drawing from a uniform distribution means that the density of birds is highest at the point of observation because no more birds are observed despite a greater area. Thus, playback volume should be randomly drawn from an exponential, log-normal, beta, or other, similar, distribution. I expect this could make the soundscape indices unnaturally sensitive to changes in abundance. This issue could be particularly problematic for common, abundant, and vocally active species like the American Robin or Eurasian Blackbird: multiple individuals are likely to be present at any given site and – under the uniform distribution scenario – their songs could artificially dominate the soundscape. I'm concerned that – if I've understood what the authors have done – this feature of the reconstruction is unrealistic and can introduce substantial bias into the analyses.

Thank you for raising this interesting and considered comment. We fully agree that, mathematically, the area surveyed on point counts increases with distance from the observer and therefore that, if uniformly distributed, the number of individuals available to be counted increases with distance. However, this does not account for a reduction in detectability with distance, such that the probability of being counted during a point count is lower if an individual is further away. The shape of detection functions for each species in each habitat is unknown but our approach for assigning playback volume (randomly sampling from a uniform distribution) effectively assumes that these two processes

(increased survey area with distance; decreased detectability with distance) cancel each other out. If anything, applying this standardised approach to assigning playback volume across all survey methods potentially inflates the relative number of more distant individuals in soundscapes constructed for sites surveyed by line transect because there isn't the increase in survey area with distance to offset the reduced detectability with distance in these data. However, over-estimating the number of more distant individuals will make the acoustic indices less sensitive to changes in community structure so our findings of pervasive, chronic declines are robust to this. Furthermore, as highlighted elsewhere, our analyses are based on standardised, site-level data so unless either the distribution of individuals relative to the observer or species' detection functions have changed systematically over time, which we have no reason to suspect has happened, this will not impact our findings.

Another quantitative remark is that US states and European countries do not seem like suitable random effects when modeling the acoustic indices (605-606). This assumes that the soundscapes across those areas have one mean value which has been repeatedly sampled, yet many US states and European countries are sufficiently large that this assumption is not accurate. Consider the soundscapes in the Italian Alps vs Sicily, or in the temperate rainforests of northern California compared to the desert of southern California – why would we expect those soundscapes to not be independent? I wonder if the authors could actually improve their explanatory power by removing these random effects; replacing them with an ecological units such as biomes/ecoregions – rather than arbitrarily defined political constructs – might be a better option.

We included the random effect of *Country* in the European models to account for differences in sampling protocol (see Supplementary table 5) and therefore, for consistency, we also included *State* as a random effect in the North American models. However, both random effect of *Country* and *State* explain very little of the variation in any of the acoustic metrics ($sd < 0.001$), so their inclusion or exclusion makes no difference to the explanatory power of the models. We have retained these random effects in our models for methodological and statistical completeness.

I think the paragraph about acoustic indices (118-132) would benefit from reorganization: I think it would make more sense to talk about the correlation between the indices and species richness in the reconstructed soundscapes before talking about the connection between indices and real-world data. This would flow better from the preceding paragraph in which the reconstructed soundscapes are discussed, so the conceptual transition is smoother.

This paragraph introduces the acoustic metrics and justifies their use in exploring changes in soundscape characteristics, in terms of their established relationships with species richness and abundance in both real-world communities (referenced papers) and simulated communities (this paper). We think it is imperative to include the results of the simulated community analyses in the main text of the paper but do not want to over-emphasise them, as the main focus of the paper is changes in the characteristics of soundscapes constructed from the monitoring data. We believe the existing structure of this paragraph delivers this balance.

In the results (158-176), there is a lot of description of change in acoustic indices but I'm left wondering what it all means in biological or experiential terms. Some translation to that perspective would be nice.

We have simplified the text in this section and included a clearer statement of the links between avian community structure, acoustic metrics and soundscape quality.

Minor Comments

Title: this is a pretty trivial point, but given that the NA-BBS data are collected in June, you've really reconstructed summer soundscapes not spring ones. Given the parallel the authors are making with Rachel Carson's famous book, leaving the title as-is is probably fine.

Original title retained

61: "data-driven" feels more like a corporate catch-phrase than something meaningful.

We feel it is important to emphasise that the historical soundscape reconstructions we analyses are directly and objectively underpinned by data rather than being, for example, some form of abstract representation. Indeed, this was a phrase suggested by the Handling Editor.

65-66: "quality" is subjective and not informative, but "acoustic diversity and intensity" are useful. Cut "quality".

Revised

75-76: the detrimental affects of school/work closures seems to be a social issue. It seems like the authors are stretching to connect a hot issue (covid-19) to their work.

We have removed the specific reference to home working and school closure but retain the more general point about the impacts of local and national lockdowns during Covid. There has been both substantial media coverage and a growing number of academic publications reporting the benefits to mental health and well-being of exposure to nature during periods of Covid-related lockdown, and vice versa. We feel this is a valuable and relatable, albeit acute, demonstration of the impacts of the more chronic nature disconnect experienced over recent decades.

77: typo? delete "necessarily"

Removed

81: the authors answer the "extent" question but only provide some speculative "implications." For this reason, I would remove implications from this sentence.

Removed

87, 95: could delete "Indeed" without affecting the meaning of the sentence

We have kept this wording as we feel it provides a stronger link to the points made in the previous sentences.

94: consider replacing "impacting" with "affecting"

Revised as suggested

108: why specify that it was the first species when there was nothing special about it being first? Just say “for each species”

This is the first step in a sequence of coded actions. To maintain clarity, we do not want to simplify/generalise our description of the methods further, especially since Reviewer 1 requested that additional details are provided.

124: delete comma after “intensity”, add comma after “therefore”

The current punctuation is correct for the purpose and format of this sentence.

125: “quality” and “value” are subjective and uninformative.

The use of these terms here is based on the findings of comparative experiments (as referenced) designed to measure these specific perceptions.

140: “quality” is finally defined here, but rather poorly. The authors write “...changes in soundscape quality, defined as a reduction in acoustic diversity and/or intensity...”. This suggests that quality = low diversity/intensity, yet this is the opposite of what the authors seem to mean. I think “quality” is quantified by acoustic diversity/intensity, with high quality = high diversity/intensity and low quality = low diversity/intensity.

Yes – quality is defined by acoustic diversity/intensity – with a reduction in quality associated with reductions in these characteristics and vice versa. We have clarified this sentence.

143-145: Is this statement about soundscape changes based on the authors’ findings, or on published research? Either a reference is missing, or the authors should combine this sentence with the following one. As written, it gives the impression that the authors have reached this conclusion before their analyses were conducted. Instead, this statement could be framed as a hypothesis which the authors set out to test.

Revised – and see ln 159.

165: “lesser”

Revised

166 and other places: “strongly” refers to force but the word is modifying a quantity, so something like “substantially” would be more appropriate

Revised as suggested

170-174: “fast” and “faster” across several sentences and it gets confusing

We have revised this part of the text to improve clarity

183: “regional, biome, and local” should be ordered by scale (one way or the other), and it isn’t really clear that this is the case

The correct ordering of regional and biome is debatable as they are measured on different scales - regional is pseudo-political/geographical whilst biome is environmental/geological. We have left this unchanged.

183-4: typo? What is “fundamental species richness and abundance?”

Revised

189-190: “will only exacerbate this process”

Revised

206-213: the recommendation that soundscape data be collected should be refined. Many hundreds of TB of soundscape data exist and are being generated each year by acoustic monitoring projects, but that data tend to be collected in relatively intact ecosystems where humans are not routinely experiencing the soundscape. My point is that this suggestion seems too general and not particularly novel. What exactly would the authors want to see from a “soundscape monitoring scheme”? Recording units across urban to rural to wilderness gradients?

We have added to this paragraph to advocate systematic recording across habitats and environmental gradients.

215: I’m in favor of the Oxford comma

As are we – this has been corrected.

216: sound is more defining than visual? I disagree

We stand by this statement, based on the reference cited here and on other references cited in the paper (e.g. Brewster & Simons (2009) *J Field Ornith* 80: 178 ; Darras et al (2019) *J Appl Ecol* 55: 2575; Franco et al (2017) *Int J Environ Res Public Health* 14: 864; Hedblom et al (2014) *Urban For Urban Green*. 13: 469; Wang & Zhao (2019) *Urban For Urban Green*. 43: 126356).

216-219: this sentence needs work. Clarify that the authors are reporting these declines, clarify that the declines are based on reconstructions (not measured declines), consider removing “concerning”

We have restructured this sentence.

226: delete “resultant”

Done

226: why is “dilution of experience” in quotes throughout the manuscript?

This is a new concept introduced in this paper and, at its first use, is formatted to match and contrast the concept of an “extinction of experience”. We have removed the quotes around subsequent uses of this term.

228-23: the authors state that “conservation policy and action need” to protect soundscapes. This raises a question about what auditory interactions with birds represent. Do people care about the sounds themselves or what they understand the sounds to represent? A conservation policy based on soundscapes alone could be to simply play recordings of complex soundscapes (e.g., reference 15). Would the authors find this solution sufficient?

As the reviewer highlights, this is a very active research area and there is still substantial work required to fully understand the mechanisms underpinning the relationships between public perceptions of, and reactions to, soundscapes with different characteristics. Ref 15

demonstrates that simple soundscape playback can deliver well-being benefits but we agree it would be very interesting to explore whether the same effects are delivered if participants know that they are being exposed to artificial soundscapes. This is beyond the scope of this paper.

All Figures: The text (axes, legends, etc.) is very small and hard to read

We would welcome guidance from the journal as to what font size is required for axes labels as we believe they are clear.

451: active voice

Not clear what is being flagged here

562: inserting a dependent clause after “that” makes the sentence hard to follow.

We have restructured this sentence to clarify

563: it seems like a comma is missing after “expected”.

As above, this sentence has been restructured

Figure S1: the color scheme is not easy to discern with such small maps; why not use the spectrum used for Fig 5?

The colour scheme for this figure has been altered as suggested.

Reviewer #3:

This paper is extremely creative and novel; I love the central idea and I think it adds a compelling new dimension to how we think about global change and the impact it is having on biodiversity, and also the human experience of nature. While there was much to like about the paper, I also thought there were a number of areas in which it could be improved.

1. I felt the topping and tailing with an extinction of experience narrative was a little incongruous, and included a number of rather speculative statements, e.g. the implication that more diverse bird song leads to greater connection to nature and therefore enhanced conservation action (for counter-examples and nuance, see e.g. Pinder et al. 2020; Oh et al 2021). [regarding the health and wellbeing outcomes of nature experiences, I agree with the authors that the evidence is clear]. As well as being rather tenuous, this narrative was distracting and is not really needed to weave the story of a degradation of natural soundscapes. In any case, the paper doesn't actually analyse anything to do with the human experience, or require recourse to any of the more speculative links between those experiences and conservation concern / action. Consider dramatically downsizing the opening and closing remarks, or reframing the first and last paragraphs to focus more simply and directly on the issue being studied.

We strongly disagree with this position, and it is in contrast to the general comments by Reviewer 2, who recognised the potentially significant implications for the human experience in their general comments. From the initial conceptualisation of this research we have focused on the quality of natural soundscapes primarily as an anthropocentric

concern, effectively framing soundscapes as an ecosystem service; that is to say, in line with the classic philosophical question about whether trees falling in a forest make a sound if no one is around to hear it, reductions in natural soundscape quality are of greatest concern if people are exposed to, and are impacted by, that change. We believe framing our research in this context provides a novel, global, and high impact narrative and have retained this general structure. As such, our opening paragraph sets out the established concept of a growing human-nature disconnect arising from an extinction of experience, and its implications for health and well-being; we do not raise consideration of implications for pro-environmental behaviour etc until the Discussion as we agree that understanding of this aspect is less developed. Much of the work around the existence and implications of an extinction of experience has been related to there being fewer and shorter opportunities to engage with nature – our paper focuses on the additional dimension of a reduction in quality of those experiences, which has received little attention to date. However, we have made edits to the Abstract to reduce the emphasis on extinction/dilution of experience and to focus on changing status of natural soundscapes.

In response to suggestions by the Handling Editor and Reviewer 2, we have also added more formal statements of our hypotheses around the impact of changes in avian community structure on soundscape characteristics. We also now include detailed analyses examining the relationship between site-level changes in species richness and abundance on site-level changes in soundscape characteristics. We believe both of these revisions further weight the balance of emphasis towards the main message of the paper – the ongoing degradation of natural soundscapes – as Reviewer 3 recommends. Furthermore, we have tempered the comment in the Discussion about the potential for reduced soundscape quality to lead to a reduction in pro-environmental behaviour, but maintain that the implications of a dilution of experience need to be explored – in the same way that this reviewer has published widely on the implications of an extinction of experience.

2. Much valuable real estate in the main paper is taken up with the results of simulations. I think much of this could be moved to supplementary information, and replaced with a deeper dive into what is driving the empirical long term changes in the acoustic indices. I really wanted to know what this soundscape-reconstruction approach is telling us that we can't get from analysing the underlying survey data – this seems to be important in justifying this approach and contributing something truly new to the literature. For example, are there non-linear dynamics, such that the soundscape suddenly falls apart as tipping points are breached, what is the relative role of species richness and abundance in explaining how rapidly the soundscape deteriorated at a site. How does composition versus species richness matter? Are increasing species (e.g. urban-adapted, invasives), contributing a particular sort of acoustic signal that is driving some of the changes.

We believe the simulations are an important component of the paper. Indeed, they clearly demonstrate both the non-linear dynamics that the reviewer refers to, with sharper drop-offs in acoustic properties at lower abundance/richness, and allow a structured assessment of specific roles of abundance and species richness in determining soundscape characteristics. However, we recognise the added value that exploring the links between changes in community composition and soundscape characteristics using “real” data brings to the paper and now included a comprehensive, site-level analysis of this using the NA-BBS and PECBMS data. As detailed in our response to Reviewer 1 on this point, these analyses

identify a strong positive relationship between changes in community structure and soundscape characteristics but also highlight the context-dependency of this; change in soundscape characteristics cannot be directly predicted from the survey data without the “translation” phase. We fully agree that the next step will be a deeper investigation of site-level changes to explore the specific mechanisms explaining this context-dependency and examining the relative role of different species and species groups in driving the observed patterns but that is beyond the scope of this paper.

3. There is no validation of the method of constructing and analysing the sound segments. Maybe I’m missing something, but I would have thought that a field validation would be rather straightforward – doing some point counts at the same time as soundscape recordings are being made, then constructing the file segments via xeno canto etc as per the methods, then comparing that statistically with the soundscape recordings. As well as comparing the indices, one could compare the number of calling events by different species etc. This might help underpin decisions about how to optimally translate the results of a survey into a sound file, and showing that the method works in a number of different settings, e.g. different habitats. This sort of underpinning validation would also give us some confidence that the indices arising from the constructed sound files actually represent a decent approximation of the soundscape at the time of the original survey.

We fully agree that this type of validation/calibration would be an interesting step to further increase the realism of soundscapes produced. However, collecting and integrating the required data would certainly not be straightforward and is not practically feasible within scope of this paper. To perform the suggested validation/calibration rigorously and robustly would i) require substantial investment in both acoustic recording units and field workers to collect sufficient paired point count data and soundscape recordings across habitats and geographical locations in North America and Europe, ii) require the soundscape construction coding to be extensively rewritten to build in species-specific inclusion rules, and iii) require all soundscapes produced for >215000 sites in each year they were surveyed to be reconstructed and reanalysed. As with NA-BBS and PECBMS surveys, fieldwork would also need to be conducted during Northern hemisphere spring/early summer to target periods of higher vocalisation during the breeding season. Most importantly, we do not believe the suggested changes would substantively alter our findings. The vast majority of initial contacts on bird surveys are aural and the sound files used to construct the soundscapes include interspersed periods of silence and vocalisation that capture the variation in song or call structure and pattern of delivery between species. We believe our objective, standardised approach that directly translates count data into soundscapes therefore both creates realistic soundscapes (an example can be found here: <https://tinyurl.com/jdcucd34>) and allows the robust analyses of changes in site-level acoustic characteristics over time.

4. The protocols for the NA-BBS and PECBMS surveys are exceedingly different, with the former being a 3-min point count of birds within 400m and the latter being a mix of "line transects, point counts or territory mapping" including all birds encountered. The authors make the brief argument that because these are internally consistent there is no problem. I’m not sure about that for two reasons. First, there could be substantial geographic heterogeneity in the types of methods employed in various parts of Europe, or temporal trends in the sorts of methods applied across the time series, leading to systematic differences in the type of data obtained. How does one turn “territory mapping” into a

simple species list used in the analysis? Second, and given that the relationship between number of indivs / species and soundscape quality is nonlinear, there might be non-linear effects on the results, because the Euro soundscapes will be far denser than North America given the more comprehensive species lists collected over a longer period. The vast differences in methods at least needs to be considered a bit more comprehensively. This comment appears to stem from a misunderstanding. Firstly, throughout our analyses, NA-BBS and PECBMS data are examined in separate models specifically because of the substantial differences in survey methods between the two schemes. This is stated in Ln 731. Secondly, our analyses of soundscape dynamics in NA-BBS and PECBMS sites are based on site-level, standardised data to account for differences in initial community composition between sites, different sampling methods between countries in Europe, and to allow comparison between the outputs of North American and European models. This is stated in Ln 727. Thirdly, we also include *Country* as a random effect in all PECBMS-based models to further account for differences in survey methods between countries in Europe (Ln 744). Note that there was no variation in the methods used to collect data within a given country during the time period examined (Ln 627). On the specific point about territory mapping, these surveys produce a data file reporting the number of territories per species at each site in each year. This format is exactly the same as data files from point count or line transect surveys.

MINOR COMMENTS

5. There is a one-size-fits-all approach to constructing the sound files (25s soundbites regardless of the type of species involved – I appreciate that includes some natural spacing of vocalisations within that segment), and I wonder instead if these could be based on estimates of calling rates, bout durations etc from the literature? The avifaunas of Europe and N America are well served with detailed handbooks containing this sort of information.

We agree that refining the soundscape construction process to incorporate species-specific vocalisation behaviour would add an additional degree of realism to the output audio files. However, this is beyond the objectives of this paper and, more importantly, is not currently feasible as there are not sufficient and consistent data for all species recorded on NA-BBS and PECBMS sites to do this accurately. Indeed, in Point 3 above, this Reviewer suggests additional fieldwork would be necessary to determine calling rates of different species. Given these constraints, we believe the standardised, objective approach we've used, with inter-species variation built into individual sound files, provides a robust representation of soundscape characteristics that allows patterns of change in acoustic properties to be detected and explored.

6. Recordings of restricted species are not directly available in xeno canto, and the fact that sites containing one or more species without a recording were removed could mean that, e.g. wilderness areas are underrepresented? This could be a systematic problem if wilderness areas are less likely to have undergone bird declines. Could we see how many (and maybe also where) the removed sites were? Did you consider using an alternative source (e.g. eBird / Macauley library collection, commercial recordings) as an alternative to fill some of these gaps?

In total, there were 11 North American species and 18 European species for which sound

recordings that met our criteria for inclusion were not available at the time of download. This led to the removal of <1.5% survey sites, distributed across 31 states, from the NA-BBS dataset and <3.5% survey sites, distributed across 9 survey schemes, from the PECBMS dataset. We have added a line detailing this to the methods (ln 646). We did not explore other sources of sound files.

7. Line 207: For interest, albeit at the risk of self-promotion (I have signed this review), I'm keen to point the authors to the Australian Acoustic Observatory (<https://acousticobservatory.org>) – a methods paper describing this network has just been accepted in MEE (Roe et al. in press).

Reference added

Peer Review Comments, second review round:

Reviewer #1 (Remarks to the Author):

The authors did a very good job addressing all my major concerns. The new analyses (i.e., regarding the changes in species richness and abundance) have strengthened the manuscript considerably and made the authors' main argument more convincing. Moreover, the methods are now clearer and easier to understand. I have no further comments. Well done!

Reviewer #2 (Remarks to the Author):

Not silent, yet: the shifting soundscapes of spring, revision 1

Previous Reviewer #2 here. My overall positive impression of the paper remains. Again, the authors used a creative approach to ask an interesting question, and I applaud them for that. My big-picture concern is that a number of the choices the authors made early in their analyses rely on unrealistic assumptions. However, based on the responses to all three reviews, it seems that the authors are not willing to make changes more substantive than redoing their linear models. The responses could be distilled to "it would be too hard to do this for all species, so we assume that the inevitable error it introduces is random." I am sympathetic to the desire to avoid fundamental changes, as rerunning the entire analysis would be extremely time-consuming. However, if the analyses are not going to be improved, the authors should acknowledge these limitations directly and limit the inferences accordingly. The conclusion should reflect the fact that the estimated reconstructions of historical soundscapes differ from the estimated reconstructions of today's soundscapes. Though all the assumptions may not be met, this raises the prospect that our baseline is shifting towards a subjectively worse place.

Examples of early choices include background noise in recordings, uniform playback volume, and constraining all communities' soundscapes to 5 minutes. I appreciate that the authors have provided responses, and I think the content of those responses needs to be incorporated into the manuscript. Each of the reviewers had major questions about the methods, and I do not think we will be the only ones. Acknowledging the limitations of the analysis pre-emptively would be valuable. Spell out for the readers what the likely outcomes of these choices are, why they do/do not matter, and how they affect the overall narrative. A dedicated subsection of the Supplementary Material would be a good place for this, and it should be referenced directly in the main text (eg, "The implications of the simplifying assumptions we made are discussed in detail in the Supplementary Material").

A major late-stage choice is failing to do any field validation (as Reviewer 3 astutely suggested). Even a case study would be compelling: choose a few locations, deploy some recorders + conduct point counts, and compare measured soundscapes to the local reconstructions – what is the correlation between actual soundscapes and their reconstructions? The authors present a straw man argument that field validation would "require" all >215000 sites to be surveyed. And to put it bluntly, even though the authors "do not believe the suggested changes would substantively alter" their findings, they should prove it.

A final point relates to the reported declines in species richness and abundance. The narrative of environmental decline is central to the narrative of this paper, and in lines 169-172 the authors report a “significant decline in total number of species and individuals counted” (Table S3, Fig S4). However, it is not clear whether this is ecologically meaningful in addition to being statistically supported. Figure 5 refers to these results, and its legend refers to the total number of species/individuals and makes no mention of data transformations. But it shows that species richness seems to change by no more or less than one species and overall abundance seemed to change by no more or less than two species. Figure S4’s y-axis labels refer to standardized totals, but the legend says “total number”. Table S3 refers to total numbers and makes no mention of standardization – which means that the parameter estimate for change in North American bird abundance is -1 individual per century! That would be a perfect case of something being statistically supported but meaningless in the real world. This confusion needs to be resolved, and Figure and Table legends need to be intelligible as standalone text. Was there meaningful change to species richness and bird abundance or was this a case of species turnover (eg, reference 42)?

Minor comments:

57: change “are also likely to be in decline” to something like “may change concomitantly”. I don’t think a soundscape itself can “decline” – just change, and whether that change is positive or negative depends on your values. That revision (or something like it) will also make it clear that you’re testing the hypothesis that bird population declines will change soundscapes (with a specific prediction that soundscape complexity will decline) – not setting out to find something you suspect.

100: didn’t the authors use count data for their reconstructions? Would the issue be more one of species richness masking changes in species composition?

Figure 1 and others: as noted before, I think increasing text size for the figures would be good. These look like base-R plots, so that should be easy to do.

Figure 4: text in the inset legends are illegibly small. The phrase “Colors indicate the size and direction of trend” doesn’t tell us which color means what, so a parenthetical clarification like “yellow is good blue is bad” is needed. Finally, I suggest removing the colored borders from the grid cells to improve visual clarity. This is particularly important for North America where the cell area to border line ratio is low relative to Europe.

617, 663: GLMM usually stands for Generalized Linear Mixed Models, and that’s the model class for which lme4 was designed. So I think “mixed” is missing here.

Reviewer #3 (Remarks to the Author):

On the framing with the extinction of experience narrative, we'll just have to agree to disagree on it. An extinction of experience is a fascinating consequence of the results that is well worth considering, but my feeling is that those consequences should be explored in the discussion, not posed a priori in the introduction given that the topic is not explored in the analyses. My thoughts seem to be aligned with those of the editor who wrote "the manuscript would benefit from framing the manuscript more on the scientific results than in the “extinction of experience” narrative as currently presented”.

I like the paper overall, and I certainly won't die in a ditch on my other comments raised, but it would be good to see some textual changes in the manuscript referring to some of these points, rather than just author-to-reviewer discussion.

Reviewer #1:

The authors did a very good job addressing all my major concerns. The new analyses (i.e., regarding the changes in species richness and abundance) have strengthened the manuscript considerably and made the authors' main argument more convincing. Moreover, the methods are now clearer and easier to understand. I have no further comments. Well done!

We thank the reviewer for their very positive response to our revisions and the manuscript.

Reviewer #2:

My big-picture concern is that a number of the choices the authors made early in their analyses rely on unrealistic assumptions. However, based on the responses to all three reviews, it seems that the authors are not willing to make changes more substantive than redoing their linear models. The responses could be distilled to "it would be too hard to do this for all species, so we assume that the inevitable error it introduces is random." I am sympathetic to the desire to avoid fundamental changes, as rerunning the entire analysis would be extremely time-consuming. However, if the analyses are not going to be improved, the authors should acknowledge these limitations directly and limit the inferences accordingly. The conclusion should reflect the fact that the estimated reconstructions of historical soundscapes differ from the estimated reconstructions of today's soundscapes. Though all the assumptions may not be met, this raises the prospect that our baseline is shifting towards a subjectively worse place. Examples of early choices include background noise in recordings, uniform playback volume, and constraining all communities' soundscapes to 5 minutes. I appreciate that the authors have provided responses, and I think the content of those responses needs to be incorporated into the manuscript. Each of the reviewers had major questions about the methods, and I do not think we will be the only ones. Acknowledging the limitations of the analysis pre-emptively would be valuable. Spell out for the readers what the likely outcomes of these choices are, why they do/do not matter, and how they affect the overall narrative. A dedicated subsection of the Supplementary Material would be a good place for this, and it should be referenced directly in the main text (eg, "The implications of the simplifying assumptions we made are discussed in detail in the Supplementary Material").

Our soundscape construction process is built on the fundamental assumption that communities containing more species/individuals produce louder and more diverse soundscapes. Whilst we accept that there will be inter-specific differences in vocalisation behaviour, we do not believe that the one individual = one contribution of song/call to a soundscape is an unrealistic baseline from which to build a standardised, objective framework for exploring temporal changes in soundscape characteristics. Furthermore, adopting a systematic approach to soundscape construction was critical to being able to explore changes in their characteristics at the scale presented here because data on e.g. call frequency/duration are not available for all species included; varying such characteristics between species would simply require other assumptions to be made. We again note that some inter-specific variation is captured within the species-specific sound files used.

We also stand by our assessment that the specific rules imposed regarding e.g. playback volume, soundscape duration, and the use of raw rather than artificially cleaned sound files, will make little difference to the overall patterns identified. We feel it is important to re-emphasise here that we report site-level trends in standardised acoustic metrics; we can see no plausible mechanism by which the construction steps flagged as unrealistic could introduce bias to our analyses and none have been suggested by the reviewer. Whilst changing soundscape and/or individual sound file duration may alter the degree of overlap in vocalisations within the soundscape, and therefore the absolute value of each acoustic metric for a given community, the simulation data already included in the manuscript suggest this will not alter the direction of change in soundscape characteristics arising from a change in community structure. For example, the change from 40 to 20 individuals (Fig 1) could also be used to reflect the change from 20 to 10 individuals if each individual vocalises twice as often, or for twice as long, as currently coded. The direction of change between these two states is consistent and, given our focus on trends in standardised, site-level data, we would therefore expect our analyses to detect equivalent patterns whatever soundscape and individual sound file durations are used. Similarly with regard concerns around the influence of background noise in some of the sound files used, we again stress that this is an entirely random factor that would, if anything, reduce the probability of detecting changes in acoustic patterns. For example, there is no reason why increasing/declining species would be more or less likely to have background noise in their sound files. As with the discussion above, whilst the presence of background noise might alter the absolute acoustic metrics, we can see no reason why it would influence patterns of change over time.

To directly demonstrate the robustness of the patterns of change in soundscape characteristics across North America and Europe presented in the paper, we have run a series of additional simulations. First, we generated a community of 10 randomly selected European bird species and specified declines in each species from 10 to 5 individuals over a six-year period. For each year we then constructed four soundscapes and extracted the associated acoustic metrics for each. The first soundscape type was built using the existing methods (5 min soundscape with playback volume sampled from uniform distribution); the second allocated playback volume sampled from a half-normal distribution; the third inserted call files into a 3min soundscapes; the fourth inserted call files into a 10min soundscape. We iterated this process for 1000 randomly sampled communities of 10 species. These simulations again show that, whilst absolute acoustic index values may differ between approaches, trends with changing community structure are broadly equivalent.

We have added substantial sections to the main text and methods section to describe the potential implications of construction methods for both absolute acoustic index values and temporal trends in those values, and to detail the additional simulations run. We also include an additional figure (Supplementary Figure 6) showing the outcomes of these additional simulations.

A major late-stage choice is failing to do any field validation (as Reviewer 3 astutely suggested). Even a case study would be compelling: choose a few locations, deploy some recorders + conduct point counts, and compare measured soundscapes to the local reconstructions – what is the correlation between actual soundscapes and their reconstructions? The authors present a straw man argument that field validation would

“require” all >215000 sites to be surveyed. And to put it bluntly, even though the authors “do not believe the suggested changes would substantively alter” their findings, they should prove it.

We did not suggest that all >215000 sites would need to be surveyed, rather that a robust validation would require comparisons to be undertaken in late spring/early summer in a range of habitats across both North America and Europe. A small-scale validation would be largely uninformative, both in terms of revising the soundscape construction process and interpreting the large-scale patterns of changing soundscape characteristics reported, because of the fine-scale temporal variation in recorded soundscape characteristics associated with a given community and location. Firstly, individual vocalisation behaviour is influenced by both intra-and inter-specific interactions and, whilst the majority of point count detections are aural, detected vocalisations range from brief contact calls to full song. Secondly, sound attenuation for a given vocalisation event will be influenced by song frequency (Hz), vegetation structure, and local environmental conditions; the contribution of a given vocalisation to a recorded soundscapes as part of the suggested validation could vary simply on the basis of wind direction relative to the recorder. Thirdly, the acoustic characteristics of any recorded soundscapes are also influenced by vocalisations from other taxa, as well as geophonic and anthropophonic noises that occur alongside avian vocalisations. The acoustic metrics extracted from a recorded soundscape would therefore not only be dependent on avian community composition, as determined by the point count, but would also vary according to each individual’s vocal response to fine-scale (temporal and spatial) factors, other biodiversity and environmental conditions. By contrast, the systematic nature of our soundscape reconstruction process effectively standardises each of these factors, generating the “average” soundscape for a given community composition. As a result, we would not necessarily anticipate the absolute acoustic metrics of recorded and constructed soundscapes for a given location at a given point in time to closely align; to overcome the stochastic nature of snapshot recorded soundscapes, an informative validation would require a large-scale comparative programme of in-field recordings and point count surveys far beyond both what the reviewer suggests and the scope of this paper. Crucially, however, the presence or absence of a strong correlation between absolute measures of acoustic indices for recorded and constructed soundscapes at a given location at a given point in time has little implication for the robustness and validity of long-term, site-level trends in acoustic conditions detected using our approach. In line with our response to the previous comment, whilst real and constructed soundscapes may sit at different starting points along the relationship between community structure and soundscape characteristics, our simulations show that the relative impact of changes in community structure will be similar across both types of soundscape.

A final point relates to the reported declines in species richness and abundance. The narrative of environmental decline is central to the narrative of this paper, and in lines 169-172 the authors report a “significant decline in total number of species and individuals counted” (Table S3, Fig S4). However, it is not clear whether this is ecologically meaningful in addition to being statistically supported. Figure 5 refers to these results, and its legend refers to the total number of species/individuals and makes no mention of data transformations. But it shows that species richness seems to change by no more or less than one species and overall abundance seemed to change by no more or less than

two species. Figure S4's y-axis labels refer to standardized totals, but the legend says "total number". Table S3 refers to total numbers and makes no mention of standardization – which means that the parameter estimate for change in North American bird abundance is -1 individual per century! That would be a perfect case of something being statistically supported but meaningless in the real world. This confusion needs to be resolved, and Figure and Table legends need to be intelligible as standalone text. Was there meaningful change to species richness and bird abundance or was this a case of species turnover (eg, reference 42)?

We apologise for any confusion here. In parallel with our analyses of trends in soundscape characteristics, our analyses of trends in total species richness and total abundance are based on site-level standardised counts. The trends presented in Figure 5 and Supplementary Figure 4 and parameter estimates in Supplementary Table 3 therefore reflect changes in SD units rather than in the actual number of species or individuals. We have further clarified Table/Figure legends and the description of models in the methods accordingly. Large scale changes in community composition and the abundance of individual species and families across North America and Europe have been widely reported elsewhere (e.g. Refs 13; 22; 42) and are not the direct focus of our study here.

Minor comments:

57: change "are also likely to be in decline" to something like "may change concomitantly". I don't think a soundscape itself can "decline" – just change, and whether that change is positive or negative depends on your values. That revision (or something like it) will also make it clear that you're testing the hypothesis that bird population declines will change soundscapes (with a specific prediction that soundscape complexity will decline) – not setting out to find something you suspect.

Text changed as requested

100: didn't the authors use count data for their reconstructions? Would the issue be more one of species richness masking changes in species composition?

Our point is that one cannot directly predict changes in soundscape characteristics from count data because of context-dependency i.e. the impact of gaining or losing a given species from a community on the associated soundscape will depend on what other species are present at that site. Instead, it is necessary to go through a "translation phase", reconstructing soundscapes and extracting acoustic indices, to measure the specific impact of site-level changes in community structure and composition on the resultant soundscapes.

Figure 1 and others: as noted before, I think increasing text size for the figures would be good. These look like base-R plots, so that should be easy to do.

The size of text of figures has been increased as requested.

Figure 4: text in the inset legends are illegibly small. The phrase "Colors indicate the size and direction of trend" doesn't tell us which color means what, so a parenthetical clarification like "yellow is good blue is bad" is needed. Finally, I suggest removing the

colored borders from the grid cells to improve visual clarity. This is particularly important for North America where the cell area to border line ratio is low relative to Europe.

Figure updated as requested

617, 663: GLMM usually stands for Generalized Linear Mixed Models, and that's the model class for which lme4 was designed. So I think "mixed" is missing here.

Text updated

Reviewer #3 (Remarks to the Author):

I like the paper overall, and I certainly won't die in a ditch on my other comments raised, but it would be good to see some textual changes in the manuscript referring to some of these points, rather than just author-to-reviewer discussion.

We have added substantial sections of new text, both in the main body of the paper and in the Methods section, that detail the discussions and our responses both here and in the previous review.

Peer Review Comments, third review round:

Reviewer #2 (Remarks to the Author):

The sensitivity analysis addressed my concerns. It might be worth emphasizing that the change in human experience would be most pronounced in the spring when birds are most vocally active.

- Reviewer #2

Reviewer #2:

The sensitivity analysis addressed my concerns. It might be worth emphasizing that the change in human experience would be most pronounced in the spring when birds are most vocally active.

We have added a comment to this effect in the Discussion (Ln 262-264).